# Global identification of functional microRNA-mRNA interactions in *Drosophila*

Hans-Hermann Wessels [1,2,3], Svetlana Lebedeva [1], Antje Hirsekorn[1], Ricardo Wurmus[1], Altuna Akalin[1], Neelanjan Mukherjee [1,4] & Uwe Ohler [1,2]

MicroRNAs (miRNAs) are key mediators of post-transcriptional gene expression silencing. So far, no comprehensive experimental annotation of functional miRNA target sites exists in *Drosophila*. Here, we generated a transcriptome-wide in vivo map of miRNA-mRNA interactions in *Drosophila melanogaster*, making use of single nucleotide resolution in Argonaute1 (AGO1) crosslinking and immunoprecipitation (CLIP) data. Absolute quantification of cellular miRNA levels presents the miRNA pool in *Drosophila* cell lines to be more diverse than previously reported. Benchmarking two CLIP approaches, we identify a similar predictive potential to unambiguously assign thousands of miRNA-mRNA pairs from AGO1 interaction data at unprecedented depth, achieving higher signal-to-noise ratios than with computational methods alone. Quantitative RNA-seq and sub-codon resolution ribosomal footprinting data upon AGO1 depletion enabled the determination of miRNA-mediated effects on target expression and translation. We thus provide the first comprehensive resource of miRNA target sites and their quantitative functional impact in *Drosophila*.

[1] Berlin Institute for Medical Systems Biology, Max-Delbrück-Center for Molecular Medicine, 13125 Berlin, Germany. [2] Department of Biology, Humboldt University, 10115 Berlin, Germany. [3]Present address: New York Genome Center, New York, NY 10013, USA. [4]Present address: Department of Biochemistry and Molecular Genetics, RNA Bioscience Initiative, University of Colorado School of Medicine, Aurora, CO 80045, USA. Correspondence and requests for materials should be addressed to U.O. (email: uwe.ohler@mdc-berlin.de)

MiRNAs are a class of ~22 nucleotide (nt) long small non-coding regulatory RNAs, involved in mRNA destabilization and translational control. In most cases, a miRNA functions as a guide directing AGO proteins via RNA-RNA-recognition to complementary target sites in the 3′ untranslated region of its target mRNA, where its repressive function gets exerted via assembly of the RNA-induced silencing complex (RISC)[1]. As miRNAs are predicted to target more than 50% of all 3′UTRs of protein coding genes in human[2] and 30% of *Drosophila* genes (TargetScanFly 6.2), either as a single miRNA or in combination, they may be the most prevalent negative regulator of posttranscriptional gene expression.

Historically, *Drosophila melanogaster* has been an important tool to study miRNAs biogenesis and function[3,4]. MiRNA gene *null* flies identified miRNAs that are critical for fly development as negative regulators of the anti-apoptosis genes *hid* (*bantam*) and *Drice* (*miR-14*)[5,6]. Many fly miRNAs exhibit spatial and temporal expression patterns and possibly spatiotemporal target gene regulation[7–9]. Advances in detecting miRNAs and their systematic annotation[9–12] have led to a current set of 466 mature *D. melanogaster* miRNAs[13].

Similar to other model organisms[14], only few fly miRNA deletions exert lethal phenotypes or strong morphological abnormalities. However, many miRNA have been found to have subtle effects[15,16], which become more pronounced when the organism is challenged. It remains difficult to describe direct organismic miRNA effects via individual targets in a quantitative manner. Here, especially human tissue culture models have greatly enhanced our understanding about miRNA function, while our understanding of fly miRNA function is lagging behind. In *Drosophila*, a recent comparative study of small RNAs across 25 cell lines suggested that the miRNA landscape in non-ovary cell lines showed little diversity and low complexity in terms of relative expression levels of individual miRNAs, which would argue against fly cell lines as a good model to study miRNA function[12].

Although knowledge of mature miRNA sequences alone have enabled the identification of physiologically relevant targets[5,17], computational methods have greatly contributed to successful miRNA target prediction, especially after recognition of the miRNA seed region (nt 2–7)[18–21]. To date, there is a plethora of computational miRNA target predictions tools, including popular approaches such as TargetScan, MIRZA, and mirSVR[2,22,23], which leverage conservation, target sequence context feature information or RNA-RNA hybridization energies and other features to improve prediction accuracy. Purely computational tools predict miRNA target sites across entire 3′UTRs, neglecting cell-type specific miRNA expression level and target site availability, which can lead to numerous and tightly spaced predictions. In vivo AGO-binding information generated from crosslinking and immunoprecipitation (CLIP) followed by sequencing (CLIP-seq or HITS-CLIP) methods has been used to greatly decrease the search space from whole 3′UTRs to about 30–40 nt per AGO footprint[24]. AGO footprints do not directly reveal the identity of the miRNA engaged, and in many cases, multiple possible miRNA seed matches overlap AGO-binding sites. However, additional anchor points, such as the coverage summit[24], but especially the presence of diagnostic events (DEs), rephrase the in vivo prediction problem to the assignment of the most plausible miRNA–mRNA pair within AGO footprints. DEs are introduced in the reverse transcription step during library generation and accumulate directly 5′ upstream of miRNA seed matches. PAR-CLIP (Photoactivatable Ribonucleoside-Enhanced Crosslinking and Immunoprecipitation) enriches for abundant nucleotide conversions (i.e., T-to-C) in the sequenced read[25] but requires RNA-labeling with photoactivatable nucleosides

(i.e., 4-Thiouridine (4SU)). For HITS-CLIP (high-throughput sequencing of RNAs isolated by crosslinking immunoprecipitation), nucleotide deletions have been mostly recognized to exhibit diagnostic potential[26,27]. iCLIP (individual-nucleotide resolution crosslinking and immunoprecipitation) on the other hand enriches for read truncations at the +1 nucleotide position of UV-crosslinked nucleotides[28]. Dedicated computational methods leverage this biochemical, single nucleotide evidence of (RNA-binding protein) RBP-RNA interaction and have improved miRNA target identification accuracy compared to solely sequence-based computational methods[29,30]. Beyond assigning miRNA seed matches in relation to single nucleotide identifiers, chimeric miRNA–mRNA reads overlapping AGO footprints can be used to unambiguously identify the interacting miRNA[31–34].

Here we describe the absolute quantification of miRNAs in *Drosophila* S2 cells and find that miRNA expression landscapes in *Drosophila* cell lines are more complex than previously reported, owing to recent technological progress in small RNA cloning. We applied both HITS-CLIP and PAR-CLIP to endogenous AGO1 protein, improving critical steps in the library cloning procedure, and compared the predictive potential of single nucleotide DEs to assign 'true' miRNA–mRNA interactions. Making use of these features, we provide the first comprehensive transcriptome-wide map of miRNA target sites in fly. Using quantitative RNA-seq and sub-codon resolution ribosomal footprinting data in response to AGO1 depletion, we further functionally evaluated and validated different types of seed matches, confirming canonical miRNA functions. We suggest that fly cell lines are suitable models to study miRNA function and provide a fully quantitative resource with comprehensive transcriptome-wide miRNA binding sites and functional readouts.

## Results

**Drosophila S2 cells show miRNA expression complexity.** In order to understand whether the low miRNA diversity previously observed in *Drosophila* cells[12] is indeed due to a low complexity in cell-type specific miRNA expression or in part due to miRNA detection limitations at that time, we generated new small RNA libraries (smRNA-seq) for *Drosophila* S2 cells using adapters with randomized ends. Fixed adapter sequences had been identified as one of the major sources of miRNA quantification biases in small RNA sequencing experiments[35,36]. Comparing mature miRNA sequences from both public and in-house S2 cell small RNA sequencing libraries we found that miRNA expression values were more evenly distributed in samples generated using randomized adapter ends (Fig. 1a and Supplementary Figure 1A). While bantam-3p alone made up ~60% of normalized miRNA reads in public smRNA-seq samples, it accounted for about ~25% miRNA reads in our new samples. Other miRNAs, such as miR-14-3p and miR-7-5p, were detected at higher frequencies. These discrepancies are likely a result of miRNA detection differences between small RNA library cloning kits rather than differences in primary miRNA expression, as normalized RNA-seq coverage was unchanged between public and in-house RNA-seq libraries (Fig. 1b). Accordingly, we found that the read sequence composition at 5′ and 3′ read ends in public samples was noticeably skewed, possibly as a consequence of non-randomized adapter ends and concomitant pronounced ligation biases (Supplementary Figure 1B). A noticeable proportion of small RNA reads from modENCODE as well as newly generated samples aligned to common *Drosophila* viral genomes. Those reads likely represent 21-nt long virus-derived siRNA and are unlikely to interfere with AGO1-mediated miRNA function (Supplementary Note 1).

Quantitative northern blot experiments confirmed that the previously lowly detected miR-14-3p was robustly detectable in

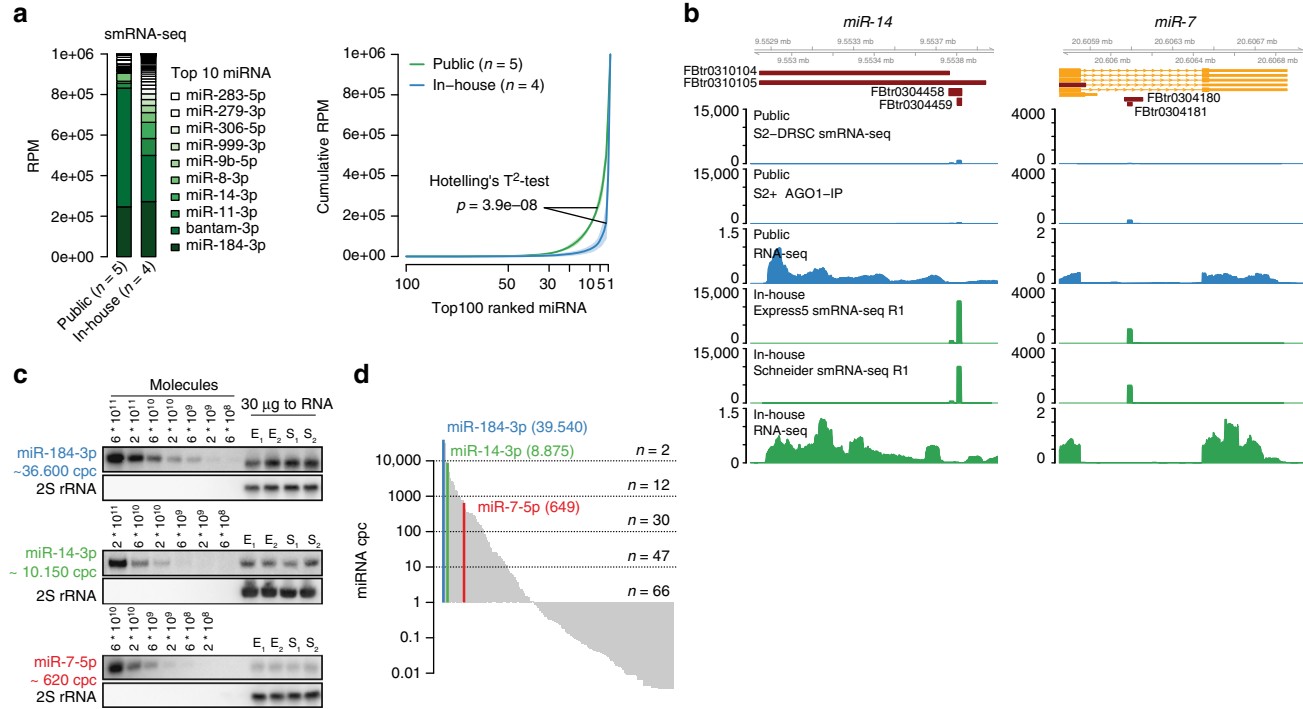

**Fig. 1** miRNA expression in *Drosophila* S2 cells is more complex than previously reported. **a** miRNA quantification in publicly available and in-house smRNA-seq samples. miRNA annotated reads were normalized to reads per million (RPM). (Left) Barplot representing the mean RPM across replicates and sorted by in-house RPM. (Right) Cumulative miRNA RPM distribution of top 100 detected and RPM-ranked miRNAs. The solid line represents the mean across libraries, shades represent the standard deviation (Supplementary Data 1). **b** Genome browser shot showing miR-14 and miR-7 reads and their respective RNA-seq coverage at miRNA loci of representative libraries normalized to total library size. Two S2 cell sub-clones have been used for new small RNA sequencing, denoted as Express5 and Schneider, respectively. **c** Quantitative miRNA northern blot for miR-184-3p, miR-14-3p, and miR-7-5p, including their experimentally determined cpc. 2S rRNA served as a loading control for total RNA samples. Source data are provided as a Source Data file. **d** Ranked distribution of fitted cpc values (Supplementary Data 1). Y-axis is in log10-scale.

S2 cells (Fig. 1c). We calculated miRNA copies per cell (cpc) for three miRNAs (miR-184-3p ~36,600 cpc; miR-14-3p ~10,150 cpc; miR-7-5p ~620 cpc) and estimated cpc for all detected miRNA in smRNA-seq samples (Fig. 1d, Supplementary Data 1). Two miRNAs (miR-184-3p and bantam-3p) were present in more than 10,000 cpc and ~30 miRNAs at more than 100 cpc. Taken together, miRNA expression levels in *Drosophila* S2 cells are more diverse than previously reported as a consequence of detection limitations.

**AGO1 HITS- and PAR-CLIP enrich for a similar set of DEs.** To identify targets of the detected miRNAs, we performed two HITS-CLIP[37] and two PAR-CLIP[25] experiments for endogenous AGO1 in S2 cells (Supplementary Figure 2A, B). We updated individual library preparation steps and performed both CLIP methods under similar conditions to be able to compare both approaches. Importantly, we replaced the RNase-T1 digestion with RNase-I digestion, which has no reported nucleotide cleavage bias, and again used 5′ and 3′ adapters with randomized ends to improve adapter ligation and help to efficiently remove PCR duplicates and, in part, sequencing errors. We sequenced all AGO1-CLIP amplicons close to estimated saturation resulting in 15,337,489 uniquely mapping reads (Supplementary Figure 2C–E). Compared with human AGO2 PAR-CLIP libraries, we observed higher relative 3′ UTR read density in fly cells (Supplementary Figure 2F). This difference may be owed to a combination of higher density of predicted miRNA target sites in fly 3′UTRs compared with human 3′UTRs[38] and possibly differences in sequencing coverage. As instructive example, we confirmed all five originally

predicted plus two additional bantam-binding sites in the *hid* 3′ UTR with AGO-binding information from HITS-CLIP and PAR-CLIP samples (Supplementary Figure 2G)[5]. Although harboring in total 45 predicted conserved and non-conserved 7/8mer seed matches for all detected miRNAs, only the predicted bantam seed matches were supported by the CLIP data.

The combination of AGO-binding information and miRNA expression levels was highly effective to pinpoint the small set of actively engaged miRNA target sites from a large compendium of computationally determined candidates (Fig. 2a, b). We analyzed all CLIP data (both HITS- and PAR-CLIP) in the same framework for more comparability (see Methods). First, we examined whether both CLIP methods would identify a similar set of AGO1-binding sites. Irreproducible discovery rate analysis indicated that both HITS-CLIP and both PAR-CLIP replicates were characterized by high peak reproducibility, while reproducibility between both CLIP methods was less pronounced (Supplementary Figure 2H). We pooled both HITS-CLIP and both PAR-CLIP replicates and selected the *n* top peaks as indicated by an IDR < 0.25 (HITS-CLIP *n* = 8971; PAR-CLIP *n* = 11,667, Supplementary Data 2 and 3) (Supplementary Figure 2H). For both CLIP methods, 3′UTR annotating peaks were enriched relative to the number of peaks expected by chance (Supplementary Figure 2I). IDR-selected AGO1-binding site positions were uniformly distributed within 3′UTRs, which is different from miRNA seed matches in human and in line with previous findings[39] (Supplementary Figure 2J).

AGO footprints do not directly reveal the identity of the bound miRNA. Several reports have exploited single nucleotide DEs introduced during library preparation as additional anchor points

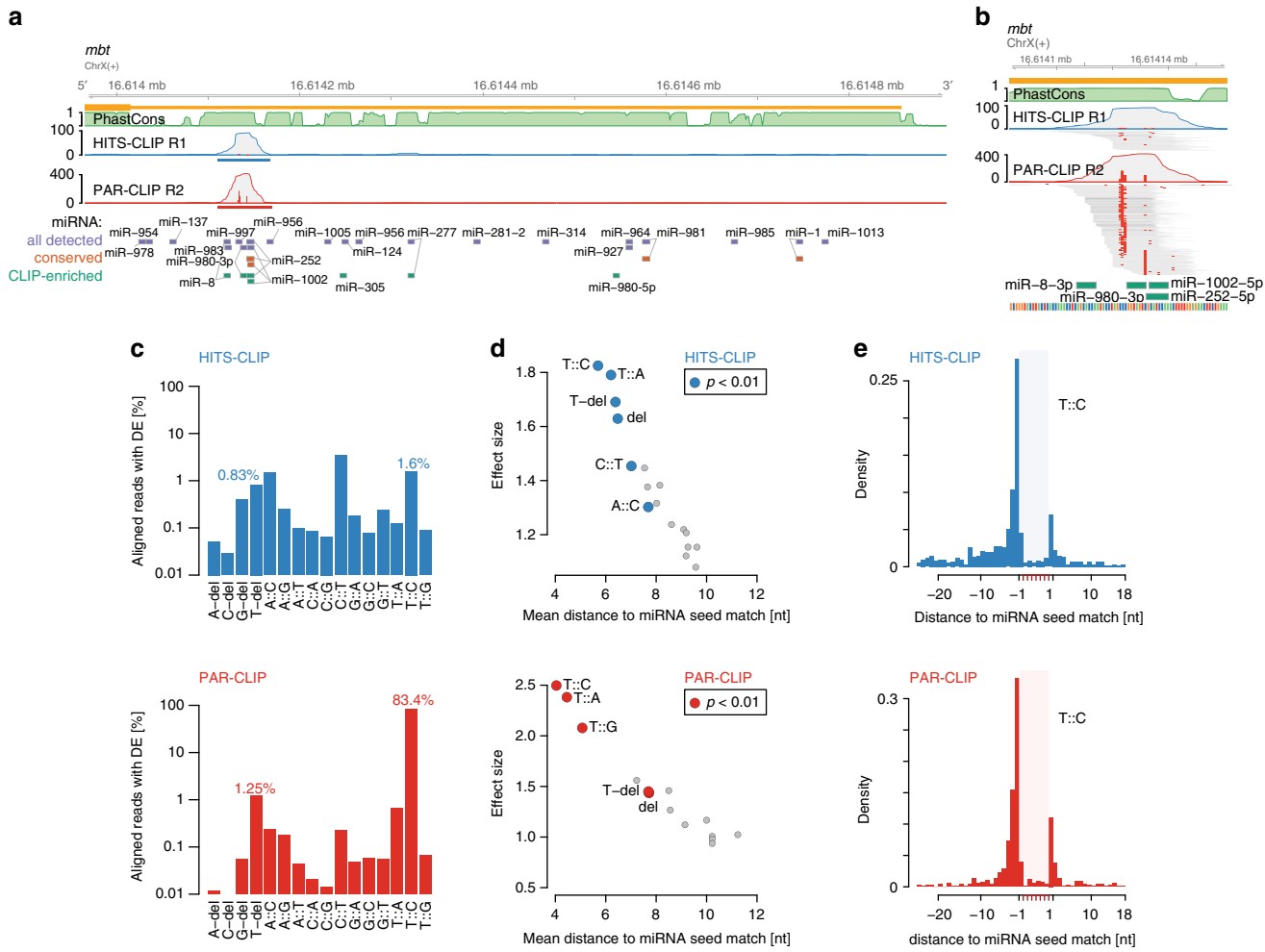

**Fig. 2** AGO1 HITS-CLIP and PAR-CLIP diagnostic event comparison. **a** Genome browser shot of the *Drosophila* gene *mbt*, depicting AGO1 HITS-CLIP (blue) and PAR-CLIP (red) coverage tracks along its 3′UTR as well as 27way PhastCons scores (green). Blue and red bars indicate IDR-selected peak calls. Below, 7mer and 8mer seed matches for all miRNA in TargetScan 6.2 (conserved and non-conserved families), conserved miRNA (predicted conserved targets), and top 59 CLIP-enriched miRNA (see Supplementary Figure 1A) are indicated (*y*-axis shows the number of detected CLIP reads). **b** Similar to (**a**), genome browser shot of HITS-CLIP and PAR-CLIP peak in *mbt* 3′UTR including alignments. Red squares in individual read alignments indicate T-to-C mismatches to the dm6 reference. Red bars within coverage tracks indicates the T-to-C conversion proportion at nucleotide resolution. Below, 7mer/8mer seed matches of CLIP-enriched miRNAs are indicated. **c** Percentages of diagnostic events relative to all uniquely aligning reads. **d** Results according to Supplementary Figure 2M. Scatterplot of mean distance to miRNA start (*x*-axis) relative to its effect size (*y*-axis). **e** T-to-C conversion example according to (**d**). Density of T-to-C conversion positional maxima relative to unique 7mer or 8mer matches in top 3000 IDR-selected 3′UTR peaks

within the AGO-binding site, which give higher resolution information about direct RBP-RNA contacts. For AGO PAR-CLIP, T-to-C conversions have been found to be diagnostic to infer miRNA seed matches 3′ downstream[25]. For AGO HITS-CLIP, nucleotide deletions were the most recognized DEs relative to miRNA seed matches[26,27]. We found that PAR-CLIP peaks showed strong positional enrichment of T-to-C conversions, which is also observed in HITS-CLIP peaks but to a lesser extent (Fig. 2b, cf.[40]).

We used the randomized adapter ends to filter aligned sequencing reads with mismatches to the reference genome to distinguish DEs introduced at crosslinked nucleotides during reverse transcription from sequencing errors. After filtering, T-to-C conversions accumulated toward the middle of mapped reads for both PAR-CLIP and HITS-CLIP samples (Supplementary Figure 2K), in contrast to previous reports for mouse AGO2 CLIP[26]. In AGO1 PAR-CLIP, more than 80% filtered uniquely aligning reads harbored T-to-C conversions (Fig. 2c). In AGO1 HITS-CLIP data, we detected more reads with T-to-C conversions (1.6%) than reads harboring T-deletions (0.83%).

In order to evaluate the diagnostic potential of all possible nucleotide conversions and deletions, we evaluated the top 3000 3′UTR peaks in detail (Fig. 2d and Supplementary Fig. 2L; see Methods). For both, PAR-CLIP and HITS-CLIP T-to-C conversions preferentially peaked 5′ proximal to unique 7mer and 8mer seed matches within AGO1 footprints (Fig. 2e). Although PAR-CLIP T-to-C conversions were by far more abundant, the less frequent conversions in HITS-CLIP can nevertheless indicate crosslinked nucleotide 5′ proximal to seed matches. In PAR-CLIP, not only T-to-C conversion, but also T-to-A, T-to-G conversions and T-deletions occur closer to seed matches than expected by chance (Fig. 2d and Supplementary Figure 2M). In HITS-CLIP, T-to-C, T-to-A conversions and T-deletions showed similar preference (Fig. 2d and Supplementary Figure 2M). About 80% of the top 3000 AGO1 HITS-CLIP 3′UTR peaks contained at least one T-to-C conversion, while T-deletions occurred in <25% and showed slightly less diagnostic potential. For both AGO1 PAR-CLIP and HITS-CLIP, crosslinked nucleotides are best indicated by T-to-V (V = A,C or G) conversions and T-deletions, though at different frequencies.

**T-centric DEs enable efficient miRNA target site prediction**. To assess the impact of T-to-V conversions together with T-deletions, we used microMUMMIE[29], a hidden Markov model that integrates CLIP binding profiles and their DEs with sequence matches to predict miRNA seed matches within AGO1-binding sites. For both CLIP methods we chose peaks with at least two DEs (hereafter referred to as cluster[41]). In AGO1 PAR-CLIP almost all IDR-selected 3′UTR peaks contain at least two T-to-C conversions ($n = 3740/3890$ 3′UTR peaks). In AGO1 HITS-CLIP more than 50% ($n = 1661/3086$ 3′UTR peaks) of the IDR-selected 3′UTR peaks were clusters based on T-to-V or T-del DE, while T-to-C conversions accounted for more clusters than T-deletions (Fig. 3a).

We ran microMUMMIE on the top 1500 PAR-CLIP clusters harboring T-to-C conversions and predicted miRNA seed matches for the miRNAs that were detected in CLIP samples relative to same number of decoy miRNAs (see Methods). CLIP samples showed a clear bimodal miRNA read distribution, suggesting that the top 59 miRNAs are actively engaged in AGO1-RISC complexes (referred to as comprehensive miRNA set) (Supplementary Figure 1A). We found that a smaller set of top 30 detected miRNAs had the best trade-off maintaining high signal-to-noise ratio (SNR), while maintaining almost maximal sensitivity (referred to as high-confidence miRNA set) (Supplementary Figure 3A, B). Comparing the predictive potential of DEs between both CLIP methods using the high-confidence miRNA set, miRNA–mRNA pairs were assigned with higher SNR at lower sensitivity values in AGO1 PAR-CLIP-derived clusters as compared with HITS-CLIP-derived clusters (Fig. 3.b, c; Supplementary Figure 3C, D). HITS-CLIP clusters may thus harbor a higher proportion of true positive miRNA seed matches compared to PAR-CLIP, but the high density of PAR-CLIP-derived DEs has a higher predictive value. In all cases, using DEs within the top 3′UTR clusters were more predictive of real miRNA seed matches than using the position of the peak summit (coverage midpoint) (Fig. 3b, c; Supplementary Figure 3C, D). While combining T-to-V or T-del DEs helped in the case of HITS-CLIP, PAR-CLIP clusters did show similar SNR and sensitivity using T-to-C conversions only. For both methods, we found a similarly strong increase of PhastCons conservation scores relative the inferred crosslinked nucleotide (Fig. 3d and Supplementary Figure 3E). In summary, we predicted miRNA seed matches for AGO1 PAR-CLIP and HITS-CLIP clusters at comparable SNRs. However, DEs were detectable as a function of sequencing depth, and their prevalence is much lower especially in AGO1 HITS-CLIP peaks with lower coverage.

**Canonical miRNA binding sites function via 3′UTR targeting**. To confirm miRNA function, we knocked down *ago1* expression using double-stranded RNA (dsRNA) mediated gene silencing and performed mRNA sequencing and ribosomal footprinting relative to control treatments (Supplementary Figure 4A, B). We calculated mRNA expression changes, changes in ribosomal footprinting, and translational efficiency (Supplementary Data 4). We assessed whether identified AGO1-binding sites in different regions of mRNAs had similar effects. For IDR-selected peaks in both CLIP methods we found that repression alleviation upon AGO1 depletion was strongest for genes bound in 3′UTRs (Supplementary Figure 4C). Changes in RNA levels and ribosomal footprinting data were concordant for the majority of AGO1-bound targets (Supplementary Figure 4D). It has been suggested previously that AGO-binding-dependent translational repression precedes RNA degradation[42] and that AGO binding in coding regions may specifically influence target gene translational efficiency[43]. In our data, only a small subset of AGO1 targets bound

in their 3′UTR were characterized by additional changes in translational efficiency that were not explained by mRNA abundance changes (Supplementary Figure 4C). We also did not observe strong changes in translational efficiency for genes targeted in coding regions relative to genes without AGO1-binding sites. However, our data was derived from 72 h dsRNA knockdown and thus may not be well suited to address preceding changes in translational efficiency.

As expression changes were most pronounced for 3′UTR bound AGO1 targets, we focused on providing a reference miRNA target site annotation to binding sites in this annotation category. Since miRNA seed match prediction on PAR-CLIP T-to-C conversions had the best SNR, and DE prevalence was much higher than in AGO1 HITS-CLIP, we reanalyzed the PAR-CLIP data using the PAR-CLIP-tailored peak caller PARalyzer[41] (Supplementary Data 5–7). First, in order to explain as many AGO1 3′UTR clusters as possible, we pooled both PAR-CLIP samples and predicted miRNA seed matches for the 59 CLIP-enriched miRNAs (referred to as comprehensive miRNA target site map; Supplementary Data 8 and 9). Similar to previous studies, not all AGO1-binding sites can be explained by a canonical miRNA seed match (up to 60%). In addition to spurious non-functional interactions in genomic crosslinking data sets, this fraction may consist at least partially of AGO1-binding sites without canonical miRNA seed match that may still be able to function in RNA-silencing (e.g., bulge sites, center sites, etc.[44–46]). However, the prevalence of such sites is still largely unclear. As an example for non-canonical miRNA binding sites, we assessed nucleation bulge site predictions for 'orphan' AGO1 3′UTR clusters. As previously reported for human AGO2 PAR-CLIP data[29], we observed a much lower, close to random signal-to-noise ratio for bulge site predictions, which altogether explained not more than 9% of all AGO1 3′UTR clusters (Supplementary Figure 4E). We found that target gene expression of genes with clusters lacking canonical seed matches (including nucleation bulges) was not noticeably different from non-targeted genes (Supplementary Figure 4F). Furthermore, slight changes may also be explained by canonical seed matches of miRNAs not included in the comprehensive miRNA set, as well as targeting in other transcript regions. Our results do not exclude the possibility that a small number of functional miRNA bulge predictions may be hidden among a much larger set of reproducible yet ineffectual sites. We have therefore added a list of reproducible miRNA bulge sites to Supplementary Data 11.

For lower ranked clusters of the pooled AGO1 PAR-CLIP data sets, prediction certainty was gradually reduced (Supplementary Figure 4G). In order to arrive at a high-confidence miRNA target site map, we predicted miRNA seed matches for the top 30 CLIP-enriched miRNA that showed good sensitivity, while maintaining high SNR on the IDR-selected peaks (referred to as high-confidence miRNA target site map; Supplementary Data 10,12 and 13). Here, we predicted miRNA seed matches on both AGO1 PAR-CLIP samples separately and kept reproducible miRNA seed match predictions. The gold standard comprises 5026 miRNA canonical seed match predictions on 2601 expressed genes (Fig. 4a). These reproducible predictions showed stronger target repression alleviation upon AGO1 knockdown than genes with non-reproducible target sites or reproducible bulge sites (Supplementary Figure 4H).

**S2 cell miRNAs reflect a terminally differentiated state**. The number of predicted targets correlated well with CLIP-derived miRNA quantification (Pearson correlation coefficient 0.47, $p = 0.012$). Accordingly, we found (the previously lowly detected and most CLIP-enriched miRNA) miR-14-3p to have the

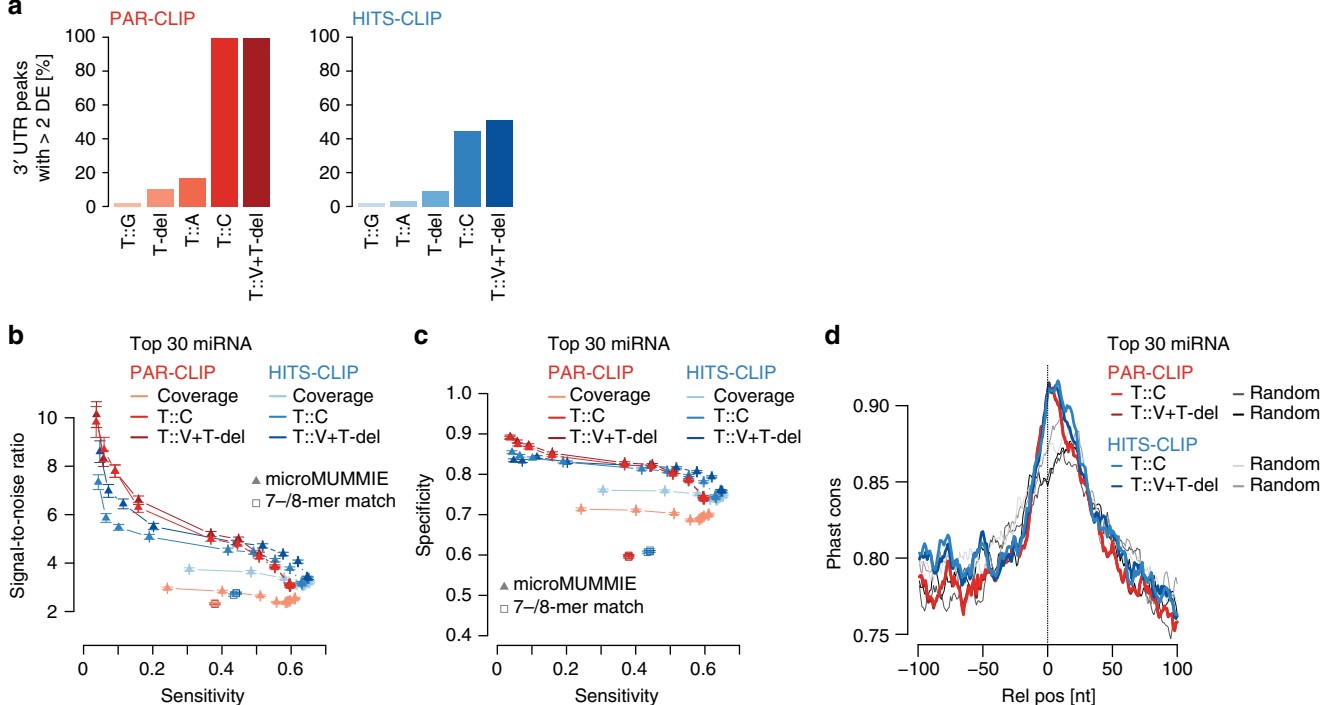

**Fig. 3** microMUMMIE assigned miRNA seed matches on PAR-CLIP and HITS-CLIP. **a** Proportion of IDR-selected peaks forming clusters (≥2 DEs per 3′UTR peak) depending on individual or combined DEs. **b** SNR estimate for HITS-CLIP and PAR-CLIP-derived DE signal for miRNA seed match predictions given the top 30 CLIP-enriched miRNAs relative to 30 shuffled decoy miRNAs. In each case, the top 1500 clusters were used. The results are depicted as mean across 100 individual shuffling experiments, with error bars representing SEM. Individual triangles indicate changes in chosen microMUMMIE variance levels. Squares show basic 7mer-A1, 7mer-m8, or 8mer-A1 matches anywhere within clusters. *X*-axis depicts sensitivity. Coverage = inferred single nucleotide peak summit position. **c** Similar to (**b**), but depicting specificity vs. sensitivity. **d** UCSC 27way PhastCons scores relative to the inferred crosslinked nucleotides for Clusters with miRNA seed match (at microMUMMIE variance 0.01; viterbi mode) prediction or a random nucleotide within the same peak

second-most reproducible miRNA target sites. On the other hand, miR-184-3p was associated with comparably few targets and did not follow this general relationship. Yet, those few targets exhibited strong repression alleviation upon AGO1 knockdown. The number of reproducible miRNA predictions had a strong cumulative effect (Fig. 4b), which was more pronounced than differences in miRNA seed match types (Fig. 4c, Supplementary Figure 4I). For some miRNAs, miRNA effects indeed increased in the order of 6mer < 7mer < 8mer (i.e., bantam-3p, miR-184-3p), but we also found examples of abundant miRNAs not showing this relationship (i.e., miR-277-3p). Individual miRNAs therefore differed from each other in target suppression strength or mode (e.g., miR-184-3p exhibited relatively strong effects on translational efficiency, while others did not) (Fig. 4d), possibly a sign of miRNA-mRNA target stoichiometry differences.

Having information about in vivo bound miRNA target sites in S2 cells provides the unique opportunity to describe the collective miRNA targetome and individual miRNA modules. We found 1237 genes being targeted by a combination of at least two miRNAs, while 1364 genes harbor one single miRNA binding site (Fig. 5a, inset). We noted that all unique miRNA target sets are larger than any miRNA pair, suggesting that no larger specific combinatorial target gene sets exist in S2 cells (Fig. 5a). To test whether the S2 cell miRNA targets address distinct biological processes, we tested for the presence of miRNA target set specific gene ontology (GO) categories. We could identify a group of strongly enriched GO terms around fly development, morphogenesis, signaling, and cell-to-cell communication for all miRNA targets combined, as well as shared across most individual miRNA target sets and differentially upregulated genes upon

AGO1-knockdown (Fig. 5b; Supplementary Data 14). We obtained similar GO-term enrichments considering only 7mer and 8mer target genes. The congruence of GO-term enrichment patterns across most individual miRNA target sets is suggesting overlapping targeted developmental processes. Indeed, calculating semantic GO-term similarities supports the notion that the majority of miRNA targets share similar GO-term enrichments (Fig. 5c).

The enriched GO-terms are suggested to be prime miRNA targets in *Drosophila*[47] in the context of fly development. We thus wanted to understand whether the miRNA targetome in fly embryo-derived S2 cells shared features with genes expressed during fly development. We noted that the 3′UTRs of genes associated with enriched GO terms were longer than the average 3′UTR length of genes expressed in S2 cells (Supplementary Figure 5A). Accordingly, miRNA target genes possessed longer 3′ UTRs compared with non-target genes in S2 cells, while mRNA expression levels were similar (Supplementary Figure 5B, C). The number of reproducible miRNA binding sites with in vivo AGO1-binding evidence correlated with the target gene 3′UTR lengths for all miRNA binding site sets (target genes at var = 0.01: $r^2 = 2.45E{-}01$, $p = 5.04E{-}161$; target genes at var = 0.5: $r^2 = 7.33E{-}02$, $p = 4.21E{-}17$). Moreover, target 3′UTRs showed increased density of predicted miRNA binding site motif occurrences for 7mer and 8mers compared with 3′UTRs of expressed non-target genes (Supplementary Figure 5D). Last, miRNA target gene 3′UTR lengths observed in S2 cells were comparable with 3′UTR length quantified from bulk embryo RNA-seq data (Supplementary Figure 5E–G). Taken together, miRNAs expressed in S2 cells target genes that associated with

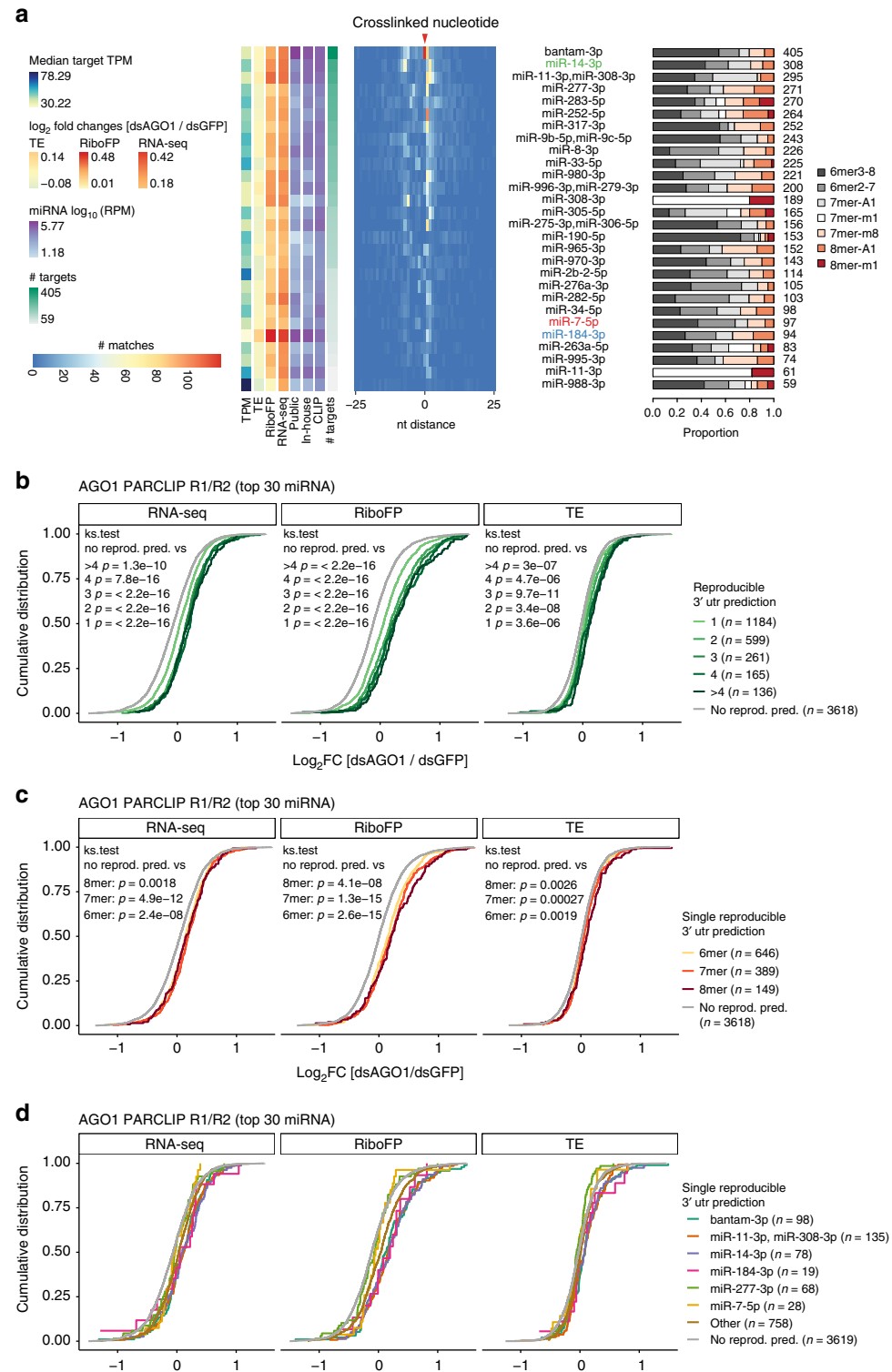

developmental cues, morphogenesis, and cell-to-cell communication. However, we cannot exclude that the targeted genes serve other non-described functions in S2 cells independent of the developmental context of the whole embryo.

## Discussion
Despite its importance as a model system, the fly community has been lacking a comprehensive, quantitative, in vivo map of *D. melanogaster* miRNA targets. To fill this gap, we describe a resource of cellular miRNA copy numbers, comprehensive miRNA target sites, as well as functional response data, including ERCC spike-in RNA-seq and matched sub-codon resolution ribosomal footprinting data utilizing randomized adapters. Here, we focused our efforts on comparing the miRNA target prediction potential for AGO1 CLIP methods, and the evaluation of miRNA function. Together, we support that at least Schneider S2 cells lines can serve as a valuable model to study fly miRNA function.

**Fig. 4** Functional evaluation of canonical miRNA seed match predictions. **a** Heatmap showing positional miRNA prediction prevalence relative to the identified crosslinked nucleotide for the top 30 CLIP-enriched miRNAs within PARalyzer-derived 3′UTR clusters. Only miRNA seed match prediction reproducible in both AGO1 PAR-CLIP replicates were considered (Supplementary Data 10). miRNAs are ranked by the number of predicted targets. The proportion of seed match types is shown on the right. On the left, the medians of steady state target expression levels (TPM), log2 fold changes of dsAGO1 vs. dsGFP treated samples for TE, RiboFP and RNA-seq are shown for all miRNA targeted genes (Supplementary Data 4), followed by the mean miRNA RPM expression levels in public and in-house smRNA-seq as well as CLIP data sets (Supplementary Data 1). Results shown were derived at microMUMMIE variance 0.01 using viterbi mode. **b** Cumulative distribution of RNA-seq, RiboFP and TE log2 fold changes for genes with 1, 2, 3, 4 or more than four reproducible miRNA seed match predictions relative to genes without reproducible predictions. P value was calculated in a two-sided Kolmogorov–Smirnov test versus genes without reproducible miRNA seed match predictions. **c** Similar as in (**b**) but isolating genes with exactly one reproducible miRNA seed match prediction stratified by 6mer, 7mer, or 8mer binding mode. **d** Similar as in (**c**) but depicting log2 fold changes for individual miRNAs (miR-184-3p, miR-14-3p, and miR-7-5p compared with three other miRNAs with the most miRNA predictions). RNA-seq, RiboFP, and TE $\log_2$ fold changes are available in Supplementary Data 4

We found the expressed miRNA pool to be more diverse than reported. T4 RNA ligases, most commonly used during small RNA cloning, were shown to have sequence biases and/or nucleic acid secondary structure hindrance in ligating single stranded RNA or DNA oligos, which can lead to miRNA mis-quantification of multiple orders of magnitude[35,48–50]. Randomizing adapter ends for miRNA cloning can efficiently reduce those biases, and results showed good agreement between complementary miRNA quantification methods[36,51]. Beyond overcoming ligation limitations, randomized adapter ends serve furthermore as unique molecular identifiers (UMIs), which help to distinguish individual ligation events from duplications introduced during PCR or sequencing. This is especially critical for low-complexity smRNA-seq libraries, where often thousands of identical reads align to only a few miRNA loci. Accordingly, we show that this new miRNA expression data is in better agreement with quantitative northern blots than previous libraries prepared without randomized adapter ends. This finding may not be limited to Drosophila S2 cells and plausibly extends to all of the 25 fly cell lines recently profiled for modENCODE[12].

For some miRNAs, detected expression levels changed drastically between public and new smRNA-seq quantification. MiR-14-3p has early on been associated with an anti-apoptotic phenotype in fly[6] and since then, it has been implicated in multiple other regulatory cues[15,52–55]. Early smRNA cloning and pyrophosphate sequencing as well as SOLiD-sequencing already indicated mir-14 to be abundant in S2 cells[9,56], but in all but one of the public smRNA libraries analyzed, mir-14 was significantly underrepresented as compared with our new smRNA-seq data (Supplementary Figure 1A). We found miR-14 levels as one of the most engaged miRNAs in AGO1-RNP complexes, targeting the second-most genes following bantam.

While we, and others[25], observed a general correlation of miRNA expression level and the number of predicted miRNA targets (Fig. 4a), miR-184 did not follow this trend. We mapped fewer target sites than expected from its expression, and its target genes were on average more strongly de-repressed upon AGO depletion and showed a relatively strong effect on translational regulation (Fig. 4a, d). MiR-184 has been previously found to be required for embryonic axis formation and has an age dependent effect on female germline development[15,57]. It has also been found responsive to high-sucrose treatment in fly and mouse as well as in diabetic mouse models[58,59], suggesting a conserved response mechanism. In both cases, the miR-184 levels are reported to drop quickly and strongly upon treatment and disease state, while miRNA expression changes are known to be normally modest. The quick drop suggests a short miRNA half-life. Given a high miRNA-mRNA target ratio in S2 cells, effective target regulation would require strong changes in miRNA levels. It seems therefore tempting to speculate that miR-184 shows common strong regulation as a result of high miR-184-target mRNA ratios in fly and mouse.

AGO-binding information greatly enhances accuracy in assigning the miRNA-mRNA gene pairs[24,25], as the search space for short miRNA seed matches is reduced dramatically from whole 3′UTRs to AGO footprints. If ambiguity remains, single nucleotide diagnostic events can be used for assigning the right miRNA[26–30]. DEs are known for all three major CLIP protocols (PAR-CLIP, HITS-CLIP, iCLIP) but so far it has been unclear how the diagnostic potential of CLIP-type specific DEs compare. In agreement with previous reports, we found T-nucleotide DEs (conversions and deletions) for miRNA seed matches located 3′-downstream for both HITS-CLIP and PAR-CLIP[40], but these DEs (especially T-to-C conversions) were much more abundant in PAR-CLIP. Contrary to previous reports[26], HITS-CLIP T-to-C conversions showed higher diagnostic potential than T-deletions, due to multiple possible reasons: (a) The original study used less stringent mapping parameters (~75% of reads contain conversions), possibly shadowing a lower fraction of informative diagnostic events. (b) Mapping of short reads including mismatches remains specifically challenging and differs across aligners[60], and the ~19× smaller fly genome (dm6 vs. mm10) implies higher mapping confidence. (c) Our use of randomized UMI adapter ends enabled us to identify and remove sequencing errors from aligned reads. A similar approach has been recently used for AGO iCLIP samples[34]. Overall, we found the combination of HITS-CLIP T-to-V conversions and T-deletions to lead to competitive SNR, sensitivity and specificity to predict miRNA seed matches, but only for the top 1500 peaks. Given that only 2.5% (1 in 40 reads) of uniquely aligned reads contain such DE, only peaks with substantial coverage can be used for this analysis, while this limitation does not exist in PAR-CLIP. For HITS-CLIP peaks without DEs, the coverage midpoint could still act as anchor point[24], but with lower SNR[29]. Importantly, our observations can be leveraged for in vivo endogenous AGO1 CLIP experiments, where the 4SU incorporation into transcripts may be difficult or impossible.

We increased the count of experimentally supported miRNA target sites in D. melanogaster from currently 12 (Diana TarBase v7.0[61]) and 150 (miRTarBase 7.0[62]), respectively, to more than 5000 reproducible sites in 3′UTRs. It is possible that miRNA targeting follows slightly different rules in different clades. For example, in C. elegans additional miRNA targeting modes were found to be more conserved than expected by chance (6mer-A1 and 8mer-1U)[38]. Moreover, miRNA seed matches in fly do not occur preferentially toward 3′UTR start and end[39,63], which is supported by our AGO1-binding data and possibly a consequence of drastically shorter 3′UTR length in flies[38]. As 3′ UTRs can undergo extensive lengthening for example in the fly nervous system and thus increase cis-regulatory 3′UTR space[64,65], this picture may depend on the tissue or differ for individual miRNAs. Furthermore, local AU-content was found to be predictive for miRNA target sites in human[2], but the 3′ UTR AU-content is higher in fly and thus may be less predictive.

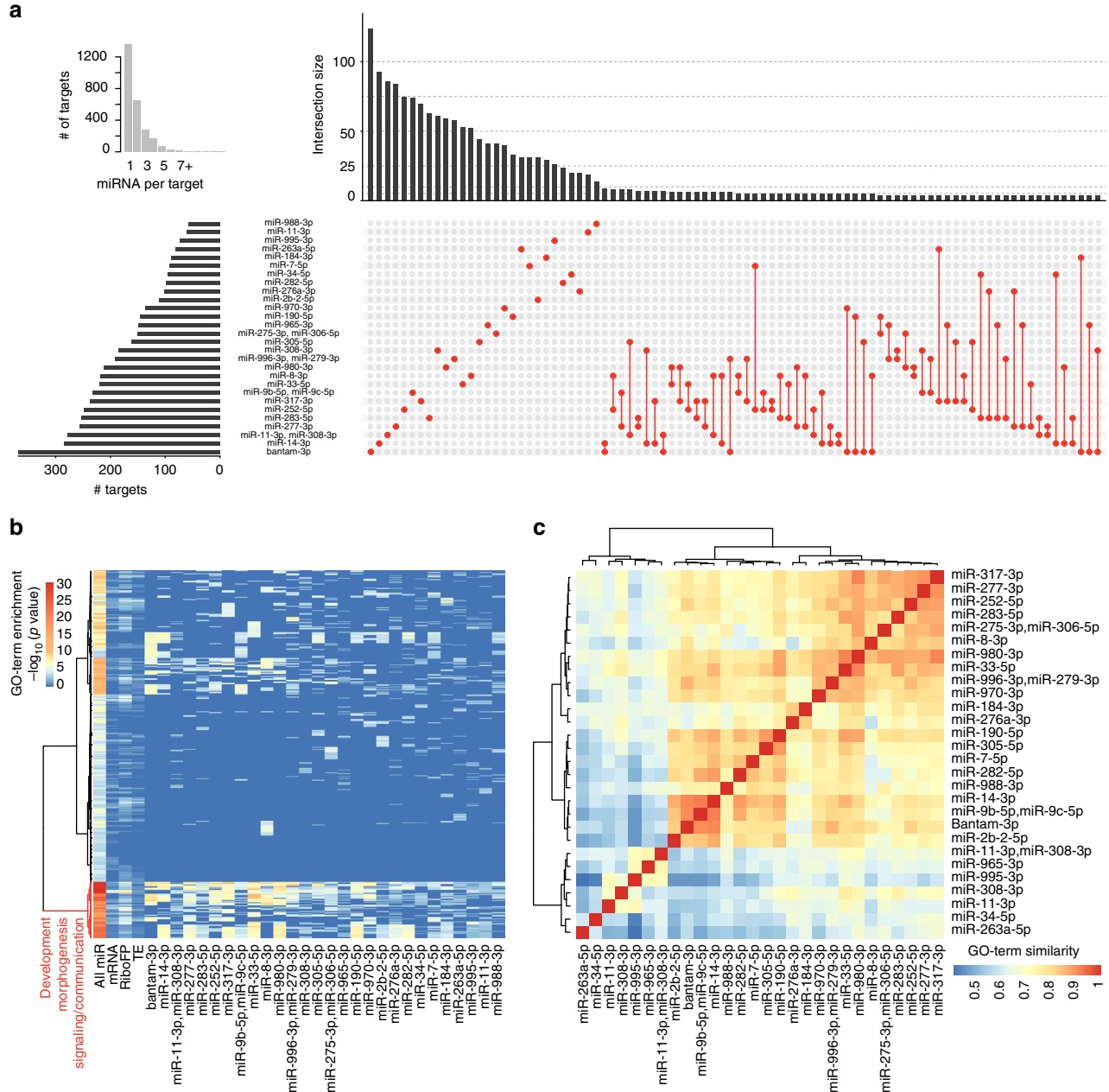

**Fig. 5** miRNAs in S2 cells collectively target genes involved in development. **a** Overview of S2 miRNA targetome. The inset on the left shows the number of detected genes with unique (=1) to up to 13 reproducible 3′UTR miRNA binding sites. Upset plot showing all possible miRNA target overlaps with minimally four shared genes (n = 88 combination > = 4 genes). Barplot on the left indicates the number of all targets per miRNA. The barplot on top indicates the size of the unique target set. The largest target gene sets exist for individual miRNAs. The largest intersect for co-targeting miRNA has a size of nine targets. The sets are indicated by red dots, connected by red lines. **b** Biological process gene ontology (GOBP) enrichment for all miRNA targets (all miR: n = 2601), top decile of genes upregulated on mRNA level upon AGO1 depletion (mRNA), top decile of genes upregulated on ribosomal footprinting level upon AGO1 depletion (RiboFP), top decile of genes upregulated on translational efficiency level upon AGO1 depletion (TE; each n = 597), and all individual miRNA target sets, relative to all genes considered during functional analysis previously (n = 5963). All significantly enriched (p < 0.001; Fisher's exact test; n = 501) GO terms for all miRNA targets were selected, merged to the corresponding enrichments in all other sets, and row-wise clustered (distance = maximum, clustering function = ward) after p value −log10-transformation, resulting in two main clusters. miRNAs are sorted by the number of targets. We did not observe enriched GO terms for individual miRNA target sets, which were not already covered by enrichments in all miR (Supplementary Data 14). **c** Pair-wise GO-term similarities using GOSemSim, for the top 100 enriched GOBP terms given p < 0.001 (Fisher's exact test), and clustered (distance = euclidean, clustering = ward)

While this study was under review TargetScan Fly v7 was released improving computational miRNA binding site predictions using reporter assays[66]. Beyond facilitating a quantitative model of miRNA targeting in fly, our comprehensive target map is thus an excellent starting point to further improve *Drosophila* target prediction.

## Methods

**miRNA quantification**. For miRNA quantification, we considered AGO1 CLIP and small RNA-seq alignments after the step of multimapper removal (see the section CLIP and smRNA-seq data processing). This was chosen for two reasons: (1) miRNAs harbor large proportions of untemplated 3′-end modification resulting in mismatches toward read ends that do not result from sequencing or adapter trimming errors. (2) Public small RNA-seq libraries (Supplementary Data 16) were

generated without introducing UMIs, which are required for UMI-based sequencing error removal.

Reads annotating to mature miRNAs were quantified and normalized to reads per million (RPM) by dividing the total number of miRNA-annotating reads and multiplication with $1 \times 10^6$. CLIP-enriched miRNAs were identified fitting a two-component mixture model[67] to the RPM-normalized and $log_{10}$-transformed miRNA counts as a mean across CLIP libraries.

To infer copies per cell (cpc) for all detected miRNAs, we first fit a linear regression model to the experimentally determined cpc and in-house smRNA-seq derived mean RPM after log-transformation. The resulting model was used to predict cpc for all detected miRNAs.

Experimentally determined miRNA cpc values fitted better with miRNA reads per million (RPM) derived from in-house smRNA-seq libraries (R-squared = 0.999, $p$ = 0.0013, residual std. er. = 52.4) than with public data sets (R-squared = 0.95, $p$ = 0.14, residual std. er. = 5810). Accordingly, fitted cpc values for all miRNAs were more coherent with in-house smRNA-seq derived RPM (in-house:R-squared = 0.989, residual std. er. = 3030; public:R-squared = 0.932, residual std. er. = 11600).

**HITS-CLIP and PAR-CLIP of endogenous AGO1 protein**. AGO1 HITS-CLIP and PAR-CLIP experiments were performed in biological replicates, originally described in Hafner et al.[25] with the following changes. Buffers were used from Huppertz et al.[68]. For AGO1 PAR-CLIP, culturing medium was supplemented with 400 μM 4SU (SIGMA #T4509), 17 h overnight before harvest. 4SU incorporation was determined to be approximately half as efficient as in HEK293T cells determined by thiol-specific biotinylation dot-blot assays as described previously[69]. Semi-adherent cells were scraped and washed in ice-cold PBS prior to 254 nm or 365 nm UV-irradiation (400 mJ/cm²), respectively. Cell pellets were snap frozen in liquid nitrogen and stored at −80 °C until further usage. For library preparation, cells were thawed on ice and lysed quickly in NP40 lysis buffer (50 mM Tris–HCl pH 7.4, 100 mM NaCl, 1% Igepal CA-630 (NP40), 0.1% SDS, 0.5% sodium deoxycholate, Complete Protease Inhibitor to a final concentration of 2 ×, RNAsin 40 U/ml lysis buffer; 1 ml lysis buffer per approximately $0.3 \times 10^9$ cells, $1.2 \times 10^9$ to $1.3 \times 10^9$ cells in total per sample). After treatment with RNaseI (1:333 v/v or 300 U/ml lysate for HITS-CLIP and 1:400 v/v or 250 U/ml lysate for PAR-CLIP samples, due to concentration differences between lysates; TurboDNase (4 U/ml) for 3 min at 37 °C and 1100 rpm). Immunoprecipitation (IP) was carried out with polyclonal AGO1-coated (Abcam #ab5070, 20 μg per sample) magnetic protein A dynabeads (Life Technologies #10002D) (100 μl) on spin-cleared cell extracts for 2 h at 4 °C.

After IP, CLIP samples were washed three times with high-salt buffer (50 mM Tris–HCl pH 7.4, 0.666 M NaCl, 1 mM EDTA, 1% Igepal CA-630 (NP40), 0.1% SDS, 0.5% sodium deoxycholate), followed by PNK-buffer washes. Samples were radioactively 5′end-labeled with γ-³²P-ATP including a subsequent addition of 1 μl high-molar ATP (100 mM) to the reaction for efficient 5′-end phosphorylation. The crosslinked protein-RNA complexes were resolved on a 4–12% Bis-Tris-polyacrylamid gel. The SDS-PAGE gel was transferred to a nitrocellulose membrane and the protein-RNA complexes migrating at an expected molecular weight were excised. RNA was isolated by Proteinase K treatment and phenol-chloroform extraction, ligated to 3′ adapter and 5′ adapter (Supplementary Data 15), reverse transcribed using Superscript III (Life Technologies #18080044), PCR-amplified (PCR cycles: HITS-CLIP 20 cycles; PAR-CLIP 19 cycles), and gel-purified. Note, after the 3′adpater ligation step, each AGO1 sample was split into approximately 19–24 nt (miRNA fraction) and 24–35-nt (target fraction) long fragments, cloned, amplified and sequenced separately. The amplicons were sequenced single-end as a multiplexed pool on HiSeq2000 (Illumina) with 51 cycles.

**CLIP and smRNA-seq data processing**. For AGO1 HITS-CLIP and PAR-CLIP libraries, sequencing reads from 19–24-nt and 24–35-nt fraction were combined before processing. For all fly CLIP libraries we quality-filtered reads using the fastx-tool kit [-q 10 -p 95] (http://hannonlab.cshl.edu/fastx_toolkit/), and adapter-trimmed using cutadapt v1.8[70] [–overlap = 3; -m 24] (Supplementary Data 15), discarding untrimmed reads. Reads were collapsed (duplicate removal) still including the four randomized nucleotides at both ends of the sequencing read. Randomized adapter ends got trimmed after read collapsing and added to the read identifier for further usage and treated as unique molecular identifiers (UMIs). As the smaller fly genome allowed higher mapping rates, we required minimally 16 nt read length. rRNA mapping reads were removed prior to aligning to the fly genome. We filtered multimapping reads and only kept the best alignment of a read if the second-best alignment had more than one mismatch more than the best alignment. Small RNA data were processed accordingly. If no randomized adapter ends (UMIs) were present, we did not apply PCR-duplicate removal. miRNA quantification on CLIP libraries was done after this processing step. Further, we filtered out all reads with mismatches relative to the genome in the first and last two nucleotides. Next, we removed reads with mismatches relative to the genome reference which were likely introduced during sequencing and thus represent sequencing errors and not diagnostic events. For this, we grouped alignments based on genomic coordinates (Chr, start, end, strand) and UMIs. In the case where alignments shared all coordinates and harbored the same UMI, while differing from each other and/or the reference sequence, we sorted by copy number (retained from read collapsing) and removed reads with relative lower copy

number and higher mismatch prevalence to the local high copy number reference read.

For the comparative CLIP analysis we called peaks using Piranha v1.2.1[71] [-s -b 20 -a 0.95 -v]. To work around Piranha's assumption that the smaller genomic coordinate is the read start irrespective of the strand, we called peaks on the read midpoints. For spliced reads, the read midpoint was assigned to the part of the read with the more extensive exon overlap. Peak reproducibility was estimated using irreproducible discovery rate (IDR)[72] on the peak read counts with an overlap ratio of 0.1. For AGO1 CLIP libraries, we chose an IDR < 0.25 for reproducing peaks between CLIP replicates and selected the top $n$ peaks (HITS-CLIP pooled $n$ = 8971; PAR-CLIP pooled $n$ = 11,667).

AGO1 PAR-CLIP data was additionally processed using PARalyzer[41] embedded in the PARpipe wrapper pipeline (https://github.com/ohlerlab/PARpipe) as described before[73]. In brief, pre-processing included the steps of adapter trimming, PCR-duplicate removal as described above. Randomized adapter nucleotides were trimmed using Flexbar (https://github.com/seqan/flexbar). Here, reads were mapped using bowtie requiring minimally 20 nt read length. Removal of rRNA reads, sequencing errors and multimapper were not applied here but left to the pipelines default setting. For group and cluster calling, PARalyzer v1.5 parameter settings were set to default except requiring minimally five unique reads to initiate a group call, while neglecting PCR-duplicate information. PARalyzer-generated clusters were filtered for T-to-C conversion specificity of at least 0.6 and higher.

**Determination of diagnostic event positional preferences**. The top 3000 3′UTR annotated IDR-selected Piranha peaks were selected for both, AGO1 HITS-CLIP and PAR-CLIP and extended on either side by 5 nt. Within peak sequences, we searched for miRNA seed matches (7mer-A1, 7mer-m8, or 8mer-A1) for the 20 most abundant miRNA in CLIP and 1000 times the same number of dinucleotide-shuffled miRNA using the TargetScan.pl script v6.1[74]. Shuffled decoy miRNAs were generated using uShuffle[75] on mature miRNA sequence and rejecting decoy sequences if they overlapped a non-shuffled (referred to as true) miRNA seed within the top 20 miRNAs. We selected peaks with exactly one seed match. Individual DE tracks at single-nucleotide resolution (i.e., all T-to-C conversion) were isolated from the CLIP alignment files and mapped relative to true miRNA or decoy miRNA seed matches in a window of ±25 nt from the genomic miRNA seed match start. For each window around a miRNA seed match the position with maximal DE occurrence was determined.

For each DE, we calculated the mean distance of maximal occurrence to the seed match start across all windows for true miRNAs and decoy miRNAs sets. Similarly, we calculated the ratio of 1/Gini-coefficient to determine positional enrichments with variable distance to miRNA seed match starts. Empirical significance was assigned with $p < 0.01$, if <1% of the 1000 individual shuffle experiments yielded lower mean distance or higher 1/Gini values than the true miRNAs. The effect size was calculated forming the ratio of the sample median of all mean distances generated by shuffling experiments and mean distance for true miRNAs.

**microMUMMIE SNR, sensitivity and specificity estimation**. miRNA target prediction evaluation for canonical miRNA seed matches was conducted in three scenarios: (1) To assess the optimal number of miRNAs to query. (2) To compare microRNA target prediction between AGO1 HITS-CLIP and PAR-CLIP. (3) To evaluate miRNA target prediction with respect to the relative rank of miRNA clusters. In each experiment, microMUMMIE was used without the option of including TargetScan-provided branch length scores shown to improve prediction accuracy in human[29]. Branch length score cut-offs for a dm6-based multiple sequence alignment have not been determined in a same way by the time this study was conducted. Available branch length score cut-offs for a dm3-based 12way multiple sequence alignment, did not improve prediction SNR.

To evaluate miRNA target prediction between AGO1 HITS-CLIP and PAR-CLIP we isolated DEs for T-to-C conversions or the combination of T-to-A, T-to-C, and T-to-G conversions as well as T-deletions (referred to as T-to-V + T-del) from fully filtered alignment files. Peaks with at least two diagnostic events were considered as clusters. Cluster boundaries were refined by trimming the edges if coverage dropped below five reads. As described in the PARalyzer method[41] we applied kernel density smoothing to the DEs within each cluster. Similarly, we determined the coverage summit. Like this, large parts of PARalyzer, including its output formats (distribution files storing smoothed DE information) were implemented in R relying Bioconductor packages[76]. For the top 1500 3′UTR clusters (width by read count) in AGO1 HITS-CLIP and PAR-CLIP, we estimated the miRNA seed match prediction accuracy using microMUMMIE, as described in the microMUMMIE methods[29]. In brief, we ran microMUMMIE using the top $n$ CLIP-enriched miRNAs plus the same number of dinucleotide-shuffled decoy miRNAs. Shuffling was done using uShuffle[75]. Decoy miRNAs were rejected, if their seed nucleotides 3–7 were overlapping with any true miRNA nucleotide 3–7 sequence to avoid overlaps to true miRNA sequences including 6mers. Only miRNA seed match predictions overlapping input clusters were retained. Predictions overlapping several transcript isoforms or miRNA seed family members were collapsed to single genomic coordinates. The signal-to-noise ratio (SNR) describes the number of true miRNA seed match predictions divided by the number of decoy miRNA seed match predictions. Sensitivity is defined as number

of clusters with at least one true miRNA seed match prediction, while specificity is defined as the ratio of true miRNA seed match predictions divided by the number of all (true and decoy miRNA) predictions. We ran microMUMMIE in viterbi mode and without conservation at 10 variance levels (var = 1.5, 1, 0.75, 0.5, 0.25, 0.2, 0.15, 0.1, 0.01, and 0.005), depicting the mean SNR, sensitivity, and specificity with its standard error of the mean (SEM) for 100 individual shuffling and training experiments.

Similarly, we estimated miRNA target prediction for AGO1 PAR-CLIP libraries processed with PARalyzer. Clusters were ranked as described above and binned into groups of 1000 clusters, before calculating SNR, sensitivity and specificity for each bin separately.

For a conservative and comprehensive set of miRNA target site predictions microMUMMIE was run on PARalyzer-derived 3′UTR clusters from both AGO1 PAR-CLIP libraries separately and only reproducible predictions were retained. MicroMUMMIE was run at six different stringency levels (variance var = 0.5, 0.25, 0.2, 0.15, 0.1, and 0.01). In the same way, we reproducible miRNA binding site maps for 5′UTR and coding regions (Supplementary Datas 13 and 12).

miRNA target prediction evaluation for non-canonical miRNA bulge matches was conducted as described previously[29] (Supplementary Data 11).

Further detailed method descriptions can be found in Supplementary Methods (Supplementary Information).

**Reporting summary**. Further information on experimental design is available in the Nature Research Reporting Summary linked to this article.

## Data availability

The AGO1 HITS- and PAR-CLIP data, small RNA-seq data, ribosomal footprinting data, and RNA-seq data generated for this study have been submitted to the NCBI Gene Expression Omnibus (GEO; http://www.ncbi.nlm.nih.gov/geo/) under accession number GSE109980. Accession numbers and summary stats of previously published data sets analyzed in this study are present in Supplementary Data 16. Final miRNA-binding site predictions are available in Supplementary Data files 8–13 and can be accessed https://dorina.mdc-berlin.de[77]. The source data underlying Fig. 1c and Supplementary Figure 4B are provided as a Source Data file. All data are available from the authors upon reasonable request.

## Code availability

The MicroMUMMIE version used in this study can be found at https://ohlerlab.mdc-berlin.de/software/microMUMMIE_Drosophila_143. Data processing descriptions for each data type can be found in its respective method section using published software. All custom code used for meta-analysis is available upon request.

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

## Acknowledgements

We gratefully acknowledge the kind gift of ERDN/SRQC small RNA spike-in reagents from the Timo Breit lab (University of Amsterdam). The anti-PABP antibody was kindly provided by Marina Chekulaeva (Max-Delbrück-Center for Molecular Medicine, Berlin). S2 cells have been a generous gift from the Robert Zinzen lab (Max-Delbrück-Center for Molecular Medicine, Berlin). We thank Maria Bikou (Max-Delbrück-Center for Molecular Medicine, Berlin) for critical reading and comments that improved the manuscript. H.H.W. and U.O. were supported in part by National Institutes of Health (NIH) grant R01GM104962 and HFSP grant RGY0093/2012.

## Author contributions

H.H.W., N.M., and U.O. conceived the experiments and computational analysis. H.H.W., S.L., and A.H. executed the experiments. H.H.W. did the computational analysis. R.W. and A.A. provided DoRiNA data upload to DoRiNA. H.H.W. and U.O. wrote the manuscript.

## Additional information

**Competing interests:** The authors declare no competing interests.

