## [Peer Review File · Nature Communications]

Reviewers' comments:

Reviewer #1 (Remarks to the Author):

In this manuscript by Wessels et al., the authors generate, validate and explore the first transcriptome wide map of microRNA (miR) binding sites in *Drosophila* cells. The authors use well-established high-throughput biochemical methods (HITS-CLIP, PAR-CLIP) to immunoprecipitate AGO1 protein and sequence the associated RNAs (miRs and mRNA target sites). The data that have been generated appear to be of relatively high quality and the data processing and analysis was done robustly with precision in calling the miR binding sites within the AGO1 bound regions (i.e. peaks). Although this work does indeed provide a useful resource for the *Drosophila* research community and likely beyond, there is very little novelty here other than the new map of miR binding sites (which in itself does have some value). Overall, this manuscript basically describes the application of established wet-lab and informatic techniques to a new system, *Drosophila*, and the results confirm much of what we already know about miR biology and functions. Also, it is a bit disappointing that this work was constrained to in vitro cell settings and not in the whole fly organisms or tissues. With the exception of identifying thousands of new target interactions (which is important), this study falls short of revealing significant new information of broader impact. For this work to be suited for a fairly broad-audience and high-impact journal like *Nature Communications*, I would expect there to be some more interesting novelty coming from the data, perhaps something new learned about miR functions. This was not the case. In the end, I believe the methods appear carefully and properly applied and the data are of quality. This paper is simply falling short on impact and novelty (based on my thorough review of other AGO CLIP papers, which have more recently advanced the methodologies, discovered new miR binding modalities (e.g. bulge sites), or investigated translationally-relevant miR interactions in primary human tissues).

Reviewer #2 (Remarks to the Author):

In this manuscript, Wessels et al. study miRNA-target interactions in *Drosophila* S2 cells through small RNAseq, CLIP, Ago1-depletion followed by ribosome footprinting. They provide a comprehensive information of miRNA target sites and analyze effects of miRNAs on target mRNA levels and their translation efficiency. While there does not appear to be a surprising new finding, the comprehensive analysis offers potentially important resources. Somewhat surprisingly, this is probably the first *Drosophila* AGO1 CLIP data so far to my knowledge. The following concerns need to be addressed before this manuscript is considered for publication in *Nature Communications*.

Major points

1) The exact origin of S2 cells is not clear (The supplemental Methods section says S2 cells from Life Technologies were used for small RNAseq, but it's unclear if the same cell line was used for the other experiments). There are many sub-clones of S2 cells and some of them show differences in miRNA expression profiles as seen in Figure S1A. While I am convinced that the current technologies allow construction of less biased small RNA libraries than the modENCODE libraries (some of them are ~10 years old), it is unclear to what extent the differences between the published libraries and the newly made libraries come from differences in the library construction methods or from biological differences between subclones. This point has to be discussed. In addition, it is a little odd not to analyze the absolute abundance of bantam-3p (By the way, in some sentences such as lines 137 and 169, it is erroneously described as bantam-5p), which is one of the main causes of the difference between the public and in-house libraries. If their S2 cells are not S2-DRSC, they should also use S2-DRSC (Available at DGRC) for the quantitative Northern blotting analysis, so the biological differences can be assessed.

2) The results of GO-term analysis using the identified miRNA targets (Figure 5B and C) led to the

identification of “a group of strongly enriched GO terms around fly development, morphogenesis, signaling and cell-to-cell communication”. I am not quite sure how to interpret the results. Firstly, I do not know results from just one cell line could tell us general features of miRNA targets. Secondly, I believe this analysis is based on the target set residing in 3' UTRs. If certain GO-terms enrich mRNAs with longer 3' UTRs, that will result in enrichment of miRNA target sites. The length of 3' UTRs should be accounted for in this analysis.

Minor point

1) Figure 1B. The mapping data show “Express 5” and “Schneider” tracks that are only explained in supplementary methods. The information should be included in the figure legends.

2) Line 261 “T-to-N conversions” may not be accurate as there wouldn't be “T-to-T” conversions. This should be “T-to-V” according to the IUPAC code.

3) “In our data, only a small subset of AGO1 targets bound in their 3'UTR were characterized by additional changes in translational efficiency that were not explained by mRNA abundance changes (Supplemental Fig. S4C).” What are those targets that show up-regulation of TE upon AGO1 depletion? Are there common features, like targeted by certain miRNAs etc.? I think Table S4 could be more useful by adding gene names (not only FBgn), targeting miRNAs, and GO category etc. Also, it would be more useful if individual supplementary tables were provided as individual files, so that readers don't need to open the ~100Mb file, for example, when they only want to see primer sequences.

Point-by-Point response

Reviewer 1)

In this manuscript by Wessels et al., the authors generate, validate and explore the first transcriptome wide map of microRNA (miR) binding sites in Drosophila cells. The authors use well-established high-throughput biochemical methods (HITS-CLIP, PAR-CLIP) to immunoprecipitate AGO1 protein and sequence the associated RNAs (miRs and mRNA target sites). The data that have been generated appear to be of relatively high quality and the data processing and analysis was done robustly with precision in calling the miR binding sites within the AGO1 bound regions (i.e. peaks).

We thank Reviewer 1 for the comments and acknowledging the thorough analysis of seed sequence-based miRNA binding site prediction presented in this manuscript.

Although this work does indeed provide a useful resource for the Drosophila research community and likely beyond, there is very little novelty here other than the new map of miR binding sites (which in itself does have some value). Overall, this manuscript basically describes the application of established wet-lab and informatic techniques to a new system, Drosophila, and the results confirm much of what we already know about miR biology and functions. Also, it is a bit disappointing that this work was constrained to in vitro cell settings and not in the whole fly organisms or tissues. With the exception of identifying thousands of new target interactions (which is important), this study falls short of revealing significant new information of broader impact. For this work to be suited for a fairly broad-audience and high-impact journal like Nature Communications, I would expect there to be some more interesting novelty coming from the data, perhaps something new learned about miR functions. This was not the case. In the end, I believe the methods appear carefully and properly applied and the data are of quality. This paper is simply falling short on impact and novelty (based on my thorough review of other AGO CLIP papers, which have more recently advanced the methodologies, discovered new miR binding modalities (e.g. bulge sites), or investigated translationally-relevant miR interactions in primary human tissues).

While we understand the individual concerns raised by the reviewer, we respectfully disagree with her/his overall assessment. Our study provides the most comprehensive *in vivo* quantitative identification of miRNA binding and effects (replicates, multiple CLIP approaches, smRNA sequencing, UMIs, model-based computational seed assignments, abundance/translation effects, etc). Due to the complexity of PAR-CLIP and other crosslinking approaches, we could not develop this setup directly *in vivo*, but our responses to rev 2 below further underline the relevance of the S2 data. As the reviewer says, we identified *thousands* of new target sites – whose biological roles should of course be ultimately backed up by *in vivo* experiments.

We agree that recent findings have proposed other miRNA binding modalities than seed matches, but we have also been highly skeptical about their prevalence. While individual ‘non-canonical’ miRNA binding sites may be functional with respect to miRNA induced gene expression silencing, the generalizability of such examples has remained controversial. For instance, the latest release of the well-known miRNA target site predictor TargetScan suggested that ‘non-canonical’ miRNA binding sites, such as bulge sites, showed no more target gene

repression than genes without any miRNA binding site re-analyzing multiple data sets ¹. In our own previous work from which we adapted the computational approach for our current study ², we had already observed a much lower, close to random signal-to-noise for bulge sites specifically. Nevertheless, ‘non-canonical’ miRNA binding sites/modalities may serve other functions than miRNA induced gene expression silencing, such as miRNA turnover ³.

In this manuscript we focused on the identification of miRNA binding sites that are functional with respect to target gene repression. In Supplementary Figure S4E, we had showed that genes that harbor ‘orphan’ AGO1 PARCLIP clusters, i.e which cannot be explained by a canonical seed match of any expressed miRNA (comprehensive top 59 miRNA set), show very little de-repression compared to genes without AGO1 binding upon AGO1 depletion from S2 cells. This suggested that on a transcriptome-wide level, most of the orphan AGO1 clusters are not functional with respect to miRNA-dependent gene silencing.

However, it was certainly important to investigate this further -- as a small subset of orphan clusters may contain functional bulge sites, we now specifically predict bulged miRNA binding sites for orphan clusters. To this end, we first tested the reliability of bulged miRNA binding site prediction using the Signal-to-Noise ratio (SNR) evaluation described in the manuscript and as used in our previous bulge analysis in human ². We distinguished between three nucleation bulge types (I, II and III) with the pivot nucleotide being opposite of miRNA nucleotide 6, 5 or 4, respectively (bulge type I equals miR-124 nucleation bulges described in ⁴).

While canonical (6-mer and up) seed match-based predictions show a clear trade-off behavior between sensitivity and SNR (increasing stringency leads to higher SNR and lower sensitivity), we did not observe a similar behavior for bulge sites irrespective of the bulge type (I, II, and III) and irrespective of the number of top peaks evaluated (1000, 5000 and 15000) (Figure 1). Furthermore, combining all bulge site types (for each set: top 1000, 5000 and 15000 3’UTR cluster) did not explain more than 9% of all 3’utr clusters (e.g. bulge type I does not exceed a sensitivity of ~0.03). This is in contrast to the findings in mouse brain HITS-CLIP data that suggested that nucleation-bulge miRNA target sites of the single abundant miRNA 124 in mouse brain can explain up to 17% (miR-124 G-bulge sites) of orphan AGO2 binding sites. Moreover, SNR is drastically lower and not much above 1, suggesting that bulge matches of true miRNAs are not found more often than control sequences close to the crosslinking sites in orphan clusters (Figure 1).

Figure 1 - SNR estimate for pooled AGO1 PARCLIP samples for miRNA seed match and bulge predictions given the top 30 CLIP-enriched miRNAs (high confidence miRNA set) relative to 30 shuffled decoy miRNAs. In each case, the top 1000, 5000 and 15000 3'UTR clusters were used. The results are depicted as mean across 100 individual shuffling experiments, with error bars representing SEM. Individual triangles indicate changes in chosen microMUMMIE variance levels. X-axis depicts sensitivity. (For bulges, only clusters were considered that did not contain a canonical seed match of a 'true' miRNA. For bulge type II (nt positions 5) evaluation and prediction, we excluded bulge seed matches to original miRNAs with the same nucleotide at positions 6, as its pairing in the nucleation step could extend one more position and thus provide for the same bulge seed match as in bulge type I. Similarly, seeds for bulge type III did not include bulge seeds that extended to bulge type II. Bulge types overlapping full length seed sequences of other miRNAs have been removed. In each case, corresponding decoy sequences got also removed)

Next, we extended the analyses presented in Supplementary Figure S4E (now S4F) of the manuscript and asked whether genes with only miRNA bulge sites in their 3'UTRs might show gene expression alleviation upon AGO1 knockdown. Those genes, targeted exclusively by bulge sites in 'orphan' clusters, did not show any evidence for gene expression alleviation upon AGO1 knockdown (Figure 2).

Figure 2 - Cumulative distribution showing mRNA-seq, RiboFP and TE log₂ fold changes for genes with 3'UTR annotating PARalyzer cluster in pooled AGO1 PARCLIP samples either with seed-based miRNA binding site predictions (= seed), with bulge-based miRNA binding site predictions (= bulge), or without (=w/o) a miRNA seed or bulge match prediction given 59 'CLIP-enriched' miRNA, relative to genes without a 3'UTR cluster. P value was calculated in a two-sided Kolmogorov-Smirnov test versus genes with no 3'UTR cluster.

We finally determined a set of reproducible bulged miRNA binding sites in 'orphan' clusters for the top 30 miRNAs quantified from CLIP data (high confidence set), analogous to the reproducible seed-based miRNA binding sites described in the manuscript. In addition to the 5026 high confidence canonical miRNA binding sites presented in the manuscript Figure 4, we found 696 reproducible bulge sites (Figure 3).

Figure 3 - Absolute numbers of reproducible miRNA binding sites at a given miRNA binding site prediction stringency, stratified by binding modality. All miRNA binding site predictions occurred in both AGO1 PARCLIP 3'UTR clusters when using the top 30 CLIP-detected miRNAs.

Even those genes that are targeted exclusively by these reproducible bulge sites do only show minor gene expression repression alleviation upon AGO1 knockdown comparable to genes with non-reproducible seed match predictions (Figure 4). However, in this analysis we cannot exclude that genes classified as targeted by reproducible bulge sites do contain canonical seed matches of miRNAs outside the set of top30 CLIP-detected miRNAs.

Figure 4 - Cumulative distribution showing mRNA-seq, RiboFP and TE log₂ fold changes for genes with reproducible seed-based miRNA binding site predictions (= reproducible_seed), with reproducible bulge-based miRNA binding site predictions (= reproducible_bulge), or with non-reproducible seed-based miRNA binding site predictions (non-reproducible) given the top30 'CLIP-enriched' miRNA, relative to genes without miRNA binding sites predictions. P value was calculated in a two-sided Kolmogorov-Smirnov test versus genes without miRNA binding sites predictions.

Moreover, we found those reproducible bulge site predictions to occur in genes with higher gene expression levels than genes with canonical miRNA binding sites (Figure 5). Furthermore, miRNA binding sites harboring bulge predictions show comparably weak coverage when normalized by gene expression level (Figure 6), suggestive of weaker miRNA binding.

Figure 5 – Boxplot showing the miRNA target gene expression levels for all genes with reproducible miRNA binding sites stratified by binding modality (according to Figure 3 – with representative prediction stringency var0.01; n represents the number of miRNA binding sites).

Figure 6 – Boxplot showing the \log_{10} -transform ratio of seed/bulge match spanning reads (coverage) and target gene TPM stratified miRNA binding modality (according to Figure 3 – with representative prediction stringency var0.01; n represents the number of miRNA binding sites).

Taken together, our SNR-based prediction assessment suggested that bulge site predictions can hardly be discriminated from control sequences, a finding that has previously been reported for human AGO2 PAR-CLIP data ². We therefore think that bulged miRNA binding sites based on AGO1 PAR-CLIP data in S2 cells may at best explain a small subset of weak ‘orphan’ clusters. Those genes show very little effect with respect to miRNA induced RNA silencing. Overall, these findings are in line with previous reports such as from the Bartel lab suggesting that nucleation bulges as a generalized binding mode may exist, but do not lead to strong measurable effects with respect to miRNA-induced gene expression silencing under physiologically relevant conditions ¹.

However, our results do not exclude the possibility that a small number of functional miRNA bulge predictions may be hidden among a much larger set of reproducible yet ineffectual sites. We have therefore added a list of reproducible miRNA bulge sites to Supplementary Data 11, edited Supplementary Figure S4E (now S4F) and S4G (now S4H) to include genes classified with miRNA bulge site predictions, and include the SNR estimates for miRNA bulge type predictions as Supplementary Figure S4E. We also comment on gene sets with bulge sites at the corresponding parts in the manuscript.

Reviewer 2)

Major point 1) The exact origin of S2 cells is not clear (The Supplementary Methods section says S2 cells from Life Technologies were used for small RNAseq, but it's unclear if the same cell line was used for the other experiments). There are many sub-clones of S2 cells and some of them show differences in miRNA expression profiles as seen in Figure S1A. While I am convinced that the current technologies allow construction of less biased small RNA libraries than the modENCODE libraries (some of them are ~10 years old), it is unclear to what extent the differences between the published libraries and the newly made libraries come from differences in the library construction methods or from biological differences between subclones. This point has to be discussed. In addition, it is a little odd not to analyze the absolute abundance of bantam-3p (By the way, in some sentences such as lines 137 and 169, it is erroneously described as bantam-5p), which is one of the main causes of the difference between the public and in-house libraries. If their S2 cells are not S2-DRSC, they should also use S2-DRSC (Available at DGRC) for the quantitative Northern blotting analysis, so the biological differences can be assessed.

We apologize for not being clear enough about the origin of the S2 cells used in this study and thank reviewer 2 for making us aware. *Drosophila* Schneider2 (S2) cells denoted as “Express5” were a generous gift from the Robert Zinzen lab (Max-Delbrueck-Center for Molecular Medicine). Those cells have been used for all experiments. S2 cells from Life technologies have been used only for Northern blotting and smallRNA-seq experiments.

We edited the Supplementary methods by adding:

“*Drosophila* Schneider2 (S2) cells were a generous gift from the Robert Zinzen lab (Max-Delbrueck-Center for Molecular Medicine).”

And by editing the sentence: “All experiments have been conducted with conditions described as above.” to “All experiments have been conducted with the S2 cell sub-clone and culturing conditions described above.”

We acknowledge the point that differences between the public and new miRNA quantification by deep sequencing could fall back to the distribution of various S2 cell sub-clones. To address this comment, we retrieved the S2-DRSC cells (stock number 181) from the *Drosophila* Genomics Resource Center (DGRC) and performed triplicate Northern blotting experiments for bantam-3p as well as miR-184-3p, comparing the two S2 cell sub-clones used in the manuscript and S2-DRSC cells.

We found that miR-184-3p was equally expressed in all three S2 cell sub clones (Figure 7 bottom). Bantam-3p was not differently expressed between the main S2 cells sub-clone used in this manuscript (Exp5) and S2-DRSC cells. However, S2 cells retrieved from Life technologies (denoted as Schneider in the manuscript) showed a marginal (+1.1 standard deviations) but significant increase in bantam-3p expression relative to both other sub-clones (Exp5 vs. DRSC $p = 0.22$; Exp5 vs. Life $p < 0.05$; Life vs. DRSC $p < 0.001$) (Figure 7 top). This marginal difference observed by bantam-3p Northern blot analysis between the two S2 cell sub-clones used in the manuscript was supported by the smallRNA-seq data presented in the manuscript (Table 1).

Figure 7 – Northern Blot analysis of bantam-3p (top) and miR-184-3p (bottom). For all three S2 cell sub-clones (Exp5 (E); Life (L) (denoted as Schneider in the manuscript) and S2-DRSC (D)) we extracted total RNA and performed northern blot analysis as described previously in the Supplementary methods of the manuscript. In each case 30µg of total RNA was loaded. 2SrRNA served as loading control. Signal densitometry was performed using FIJI. The tables below the blot results show the raw densitometry signal for miRNA and 2S rRNA. We calculated a loading factor based on relative 2S rRNA signal and multiplied the raw miRNA signal for normalization. The Zscore indicates the relative expression difference between each S2 cell sub-clone. P values have been calculated between the replicate Zscores of the S2-cell sub-clones. R1, R2, R3 denote the replicates.

Table 1: Normalized smallRNA-seq miRNA counts (RPM) for miR-184-3p and bantam-3p. Excerpt from Supplementary Table S1.

	Express5 R1	Express5 R2	Schneider (Life) R1	Schneider (Life) R2
bantam-3p	217425.9	189008.8	263501.7	241255.1
miR-184-3p	265460	267833.8	259693.8	294649.5

Taken together, we believe that there is strong evidence that the differences in miRNA quantification between previous public smallRNA-seq data and our new smallRNA-seq data presented in Figure 1 are based on improved sequencing technologies. Differences in miRNA expression between S2 cell sub-clones may exist, but may have only marginally contributed to the miRNA quantification in Supplementary Data 1.

Major point 2) The results of GO-term analysis using the identified miRNA targets (Figure 5B and C) led to the identification of “a group of strongly enriched GO terms around fly development, morphogenesis, signaling and cell-to-cell communication”. I am not quite sure how to interpret the results. Firstly, I do not know results from just one cell line could tell us general features of miRNA targets. Secondly, I believe this analysis is based on the target set residing in 3’ UTRs. If certain GO-terms enrich mRNAs with longer 3’ UTRs, that will result in enrichment of miRNA target sites. The length of 3’ UTRs should be accounted for in this analysis.

We thank reviewer 2 for highlighting the importance of gene/transcript-specific 3’UTR length with respect to possessing/acquiring miRNA binding sites and the possibility for UTR-length-dependent GO-term enrichment. Earlier work has highlighted that for example in developing organisms, such as *Drosophila melanogaster* embryos, 3’UTR lengths can undergo drastic changes over time within one tissue and/or in a tissue specific manner ^{5,6}. As GO-terms are typically assigned to a gene and not to individual isoforms, some may be incompletely assigned as genes can carry out various function given a certain 3’UTR ⁷.

To address the reviewer’s comment, we first calculated the mean gene-wise 3’UTR length and found that targeted genes have longer 3’UTRs than genes without reproducible miRNA binding site, while showing comparable mRNA expression levels (Figure 8). We observed a significant correlation between 3’UTR length and the number of reproducible miRNA binding sites across all miRNA prediction stringency ($\text{var}0.01 < \dots < \text{var}0.5$) thresholds (Table 2). Moreover, we found that the reproducible miRNA binding sites analyzed in Figure 5 distributed evenly across 3’UTRs, suggesting no present bias for miRNA target sites found at 3’UTR starts (Figure 9).

Figure 8 – miRNA target gene expression and target gene 3'UTR length. Boxplots showing the target gene expression levels (upper panel) and mean gene-wise 3'UTR lengths of miRNA targets at different microMUMMIE stringency cutoffs. Transcript 3'UTR lengths were extracted from ensemble v81 GTF using the Bioconductor package *Genomic Features*⁸. Transcript isoform percentage was estimated using *RSEM*⁹ for wildtype S2 cell total RNA-seq samples generated for this study. The mean gene 3'UTR length was calculated by multiplying transcript 3'UTR length times estimated isoform percentage by 100. We considered only genes that have been reliably detected and have been used to calculate mRNA, ribosomal profiling and translational efficiency changes presented in ($n=5963$, in Supplementary Data 4).

Table 2: Correlation between the number of reproducible miRNA binding sites and the mean gene 3'UTR length

	var0.01	var0.1	var0.15	var0.2	var0.25	var0.5
adj.r ²	2.45E-01	2.32E-01	2.13E-01	1.92E-01	1.64E-01	7.33E-02
P-value	5.04E-161	9.48E-141	7.06E-119	1.32E-96	1.81E-71	4.21E-17

Figure 9 - Relative distribution of reproducible miRNA binding sites in targeted 3'UTRs. We used the high confidence miRNA binding sites set analyzed in Figures 4 and 5 (Supplementary Data 10 – microMUMMIE var0.01). (For spatial analysis we used Spatial.pl and Spatial.R including transcript isoform tracking of the PARCLIP data analysis pipeline PARpipe. The red line represents the observed distribution of reproducible miRNA binding sites, relative to randomized locations positioning (grey and blue)).

Next, we assessed whether enriched GO-term categories contain genes with long 3'UTRs, given genes expressed and targeted in S2 cells. For all enriched GO-terms (n=501, in Supplementary Data 14, and Figure 5B) we queried the mean 3'UTR length of miRNA-targeted genes. We compared the median 3'UTR length of each GO-category to the total 3'UTR length distribution of all reliably detected genes in our study. We found that almost all enriched GO-categories contain genes sets that (in S2 cells) show comparably longer 3'UTRs (Figure 10). This is in concordance with the observation that expressed genes targeted by miRNAs in S2 cells harbor longer 3'UTRs (Figure 8). We did not see a correlation (p-value: 0.703) between the significance of an enriched GO-term and the median 3'UTR length of the associated gene sets.

Figure 10 – Histogram of median 3'UTR length of genes assigned to enriched GO-categories (n=501). For each Gene ontology biological process presented in (n=501, in Supplementary Data 14, and Figure 5B), we extracted the associated gene identifiers and determined the median of each 3'UTR length distribution. The medians of each of the 501 distributions exceeds in most cases the average 3'UTR length of all reliably detected genes in S2 cells.

As proliferating cell lines have been shown to often display a significantly different 3'UTR usage compared to non-proliferating tissues and cells in human¹⁰, we finally assessed how S2 cell 3'UTR lengths compare to 3'UTR lengths in the developing fly. We analyzed the modENCODE developmental transcriptome¹¹ and calculated gene expression counts and mean gene-wise 3'UTR usage as described above. Principal component analysis (PCA) of gene expression counts positioned S2 cells more closely to whole embryo samples than to other developmental stages PCA (Figure 11) as expected given their late embryonic origin¹². Moreover, 3'UTR length distributions were merely not significantly different between expressed genes in embryonic samples and genes expressed in S2 cells when comparing miRNA-targeted genes only (Figure 12). And on gene level, 3'UTR lengths correlated significantly with 3'UTR length in S2 cells (adj. R^2 between 0.75 – 0.8) (Figure 13).

Figure 11 – Principal Component Analysis (PCA) of mRNA gene expression counts of whole organism samples and in-house S2 cells. Whole organism gene expression data has been obtained from modENCODE¹¹, while only considering single stranded, non-stranded polyA selected data from the Gravelly lab. RSEM-derived gene expression counts were rlog-transformed using DESeq2¹³. PCA-analysis was done on the 1000 most variant genes (given TMP and read counts > 5). Replicates have been merged by calculating mean expression per gene.

Figure 12 – 3'UTR length distributions of genes with reproducible miRNA target sites in S2 cells (TPM and read count > 5). Asterisks indicated significance between 3'UTR length of a given samples compared to 3'UTR length distributions overserved for the sample genes in S2 cells (two sided Kolmogorov-Smirnoff test, Signif. codes: 0 '***' 0.001 '**' 0.01 '*' 0.05).

Figure 13 – Heatmap depicting the $adj. R^2$ of gene-wise comparison of 3'UTR in embryonic samples compared to 3'UTR length in S2 samples for genes targeted by reproducible miRNA binding sites at different microMUMMIE stringency cutoffs. (Scale color code = $adj. R^2$)

Taken together, we see that genes with reproducible miRNA binding sites in S2 cells tend to harbor longer UTRs and that 3'UTR length correlates with the number of empirical miRNA binding sites observed in S2 cells. Enriched GO-terms presented in Figure 5B do indeed show enrichments for genes with longer 3'UTRs. Moreover, we believe that the overall 3'UTR repertoire of genes expressed and miRNA-targeted in S2 cells is not significantly different from

the 3'UTR usage in bulk embryonic samples. We cannot exclude that individual genes show strong UTR dependent expression/targeting differences between whole organismic samples (especially individual cell embryonic cell populations) and S2 cells, but we believe that the 3'UTR length may only have a minor to negligible influence on our result presented in Figure 5.

Minor point 1) Figure 1B. The mapping data show “Express 5” and “Schneider” tracks that are only explained in Supplementary methods. The information should be included in the figure legends.

We edited the Figure 1B legend by adding:

“Two S2 cell sub-clones have been used for new small RNA sequencing, denoted as Express5 and Schneider, respectively.”

Minor point 2) Line 261 “T-to-N conversions” may not be accurate as there wouldn't be “T-to-T” conversions. This should be “T-to-V” according to the IUPAC code.

We thank reviewer 2 for making us aware of this inaccuracy. We edited T-to-N conversions to T-to-V conversions throughout the manuscript.

Minor point 3) “In our data, only a small subset of AGO1 targets bound in their 3'UTR were characterized by additional changes in translational efficiency that were not explained by mRNA abundance changes (Supplementary Fig. S4C).” What are those targets that show up-regulation of TE upon AGO1 depletion? Are there common features, like targeted by certain miRNAs etc.? I think Table S4 could be more useful by adding gene names (not only FBgn), targeting miRNAs, and GO category etc. Also, it would be more useful if individual Supplementary tables were provided as individual files, so that readers don't need to open the ~100Mb file, for example, when they only want to see primer sequences.

We have added the information that miR-184-3p shows the strongest suppressive effects on target gene translational repression (compare Figure 4D).

Also, we thank reviewer 2 for pointing out current limitations in the representation of our resource data. In addition to depositing the data on GEO and in the Supplement, we are uploading all peaks, clusters and miRNA seed match predictions to the RNA regulatory interaction database doRiNA (<https://dorina.mdc-berlin.de>) to improve data availability. The data will be deposited by the time of final submission. Moreover, we included gene names, miRNA name and miRNA target site count for the high confidence miRNA set in Supplementary Data 4.

We found the addition of further gene-level features (e.g. GO-terms) in a single table to be too complex given multiple assignments per gene. We therefore kept this information separate.

We also understand, that sharing all data in a single file is not convenient. We now added all Supplementary Data files as individual tables.

References

1. Agarwal, V., Bell, G. W., Nam, J.-W. & Bartel, D. P. Predicting effective microRNA target sites in mammalian mRNAs. *Elife* **4**, 1–38 (2015).
2. Majoros, W. H. *et al.* microRNA target site identification by integrating sequence and binding information. *Nat. Methods* **10**, 630–633 (2013).
3. Rügger, S. & Großhans, H. MicroRNA turnover: When, how, and why. *Trends Biochem. Sci.* **37**, 436–446 (2012).
4. Chi, S. W., Hannon, G. J. & Darnell, R. B. An alternative mode of microRNA target recognition. *Nat. Struct. Mol. Biol.* **19**, 321–7 (2012).
5. Hilgers, V. *et al.* Neural-specific elongation of 3' UTRs during *Drosophila* development. *Proc. Natl. Acad. Sci. U. S. A.* **108**, 15864–9 (2011).
6. Smibert, P. *et al.* Global patterns of tissue-specific alternative polyadenylation in *Drosophila*. *Cell Rep.* **1**, 277–89 (2012).
7. Berkovits, B. D. & Mayr, C. Alternative 3' UTRs act as scaffolds to regulate membrane protein localization. *Nature* **522**, 363–367 (2015).
8. Lawrence, M. *et al.* Software for Computing and Annotating Genomic Ranges. *PLoS Comput. Biol.* **9**, 1–10 (2013).
9. Li, B. & Dewey, C. N. RSEM: accurate transcript quantification from RNA-Seq data with or without a reference genome. *Genome Biol.* **12**, 1–16 (2011).
10. Mayr, C. & Bartel, D. P. Widespread Shortening of 3' UTRs by Alternative Cleavage and Polyadenylation Activates Oncogenes in Cancer Cells. *Cell* **138**, 673–684 (2009).
11. Graveley, B. R. *et al.* The developmental transcriptome of *Drosophila melanogaster*. *Nature* **471**, 473–9 (2011).
12. Schneider, I. Cell lines derived from late embryonic stages of *Drosophila melanogaster*. *J. Embryol. Exp. Morphol.* **27**, 353–65 (1972).
13. Love, M. I., Huber, W. & Anders, S. Moderated estimation of fold change and dispersion for RNA-seq data with DESeq2. *Genome Biol.* **15**, 550 (2014).

Reviewers' comments:

Reviewer #2 (Remarks to the Author):

In the rebuttal letter, the authors satisfactorily addressed most of the points raised in my last review. I only have a minor point as described below, which I believe could be addressed.

I think my comments about the 3'UTR length may have been partially misunderstood. Although I acknowledge that the authors performed solid analysis for the rebuttal letter (Figure 8-13), I didn't feel my point was really addressed. My point was that we don't know if the observed GO-term enrichment was a mere result of enriching longer 3'UTRs in the set of genes with miRNA target sites or the results imply any importance of miRNAs in the biological processes. For example, if one makes a group of genes based on the presence of a hypothetical sequence motif with no functionality that is distributed randomly throughout the 3'UTR-ome (let's say, like a rare restriction enzyme site), that may end up enriching longer 3'UTRs in the group, simply because longer 3'UTRs have higher probability to contain the motif. In the end, it is possible that something with no functionality could end up enriching some GO-terms that are associated with long UTRs.

For this, if the density of miRNA target sites (the number of target sites per kb, not the number of target sites per transcript) is higher in the sets of genes with the particular GO-terms discussed in Figure 5, I think that kind of results might be better evidence.

Of course, GO-term analysis only tells us correlation, and I find that the authors are careful in phrasing not to imply any functional importance of miRNAs in the enriched biological processes, but I think this may not be enough for general audience. If there is no strong evidence excluding the possibility that this is a mere result of enriching long 3'UTR genes, the authors should let readers know the potential caveat. In fact, it is a little worrisome that the group of genes showing up-regulation upon AGO1 depletion ("mRNA" group in Figure 5B) does not seem to show particularly stronger enrichment of the same set of GO-terms and may not further support the functional relevance of observations discussed throughout Figure 5.

Another suggestion is to consider including some of the results presented in Figures 7-13 of the rebuttal letter as supplementary figures and briefly describing them. The information seems useful for the research community.

Reviewer #2 (Remarks to the Author):

In the rebuttal letter, the authors satisfactorily addressed most of the points raised in my last review. I only have a minor point as described below, which I believe could be addressed.

I think my comments about the 3'UTR length may have been partially misunderstood. Although I acknowledge that the authors performed solid analysis for the rebuttal letter (Figure 8-13), I didn't feel my point was really addressed. My point was that we don't know if the observed GO-term enrichment was a mere result of enriching longer 3'UTRs in the set of genes with miRNA target sites or the results imply any importance of miRNAs in the biological processes. For example, if one makes a group of genes based on the presence of a hypothetical sequence motif with no functionality that is distributed randomly throughout the 3'UTR-ome (let's say, like a rare restriction enzyme site), that may end up enriching longer 3'UTRs in the group, simply because longer 3'UTRs have higher probability to contain the motif. In the end, it is possible that something with no functionality could end up enriching some GO-terms that are associated with long UTRs.

For this, if the density of miRNA target sites (the number of target sites per kb, not the number of target sites per transcript) is higher in the sets of genes with the particular GO-terms discussed in Figure 5, I think that kind of results might be better evidence. [...]

We apologize for not addressing the reviewer 2 remark concerning the connection between GO-term enrichments and 3'UTR lengths to a satisfactory degree. We thank reviewer 2 for stating the remark more precisely.

We have now estimated the miRNA motif density for the top 30 miRNAs used in the analyses presented in Figures 4 and 5 for targets and non-targets. For this, we selected for each gene the transcript with the highest isoform percentage determined from the RNA-seq data using RSEM. (The Figures 7-13 in the previous point-by-point response were based on the mean 3'UTR length per gene, taking into account the isoform percentage. All conclusions presented previously remained the same when considering only the transcript isoform with the highest isoform percentage.) We extracted the corresponding 3'UTR sequences and quantified the miRNA motif density of the top 30 detected miRNA as well as 100x individual di-nucleotide shuffled sets of decoy miRNA sequences irrespective of AGO1-binding data. We considered all genes with miRNA binding sites as 'targets' and all genes reliably detected and used in the RNA-seq and Ribo-seq analysis, but without AGO1-binding as 'non-targets'. We arrived at the miRNA motif density by dividing the number of motif hits by the total 3'UTR length of the respective gene set.

We found that the total putative miRNA motif density is on-par for AGO1-bound target genes (*red point*) and expressed non-target genes (*grey point*) when considering all miRNA motif types (Figure 1). Stratified by miRNA motif type we find 8mers and 7mers enriched in target genes,

while 6mers are slightly depleted. In each case, the putative miRNA motif hit densities of true miRNAs were relatively depleted compared to the motif density achieved for shuffled decoy sequences in both, targets and non-targets (red and grey distributions; black bar indicates the distribution mean), indicating that miRNA targeted sequences are relatively depleted in 3'UTRs. This trend remains for individual 3'UTR level (Figure 2). However, it is important to note that the motif density is not directly indicative of AGO1 binding. AGO1-targets and non-targets are expressed at comparable levels (shown in the previous point-by-point response), but although the putative seed match motifs are present in non-targets, they did not get recognized by AGO1 *in vivo*. Thus, AGO1 binding data is more informative than miRNA motif densities. Given the increased density of higher order miRNA seed match types (8mers and 7mers), we tested for enriched GO-terms only considering genes targeted by reproducible 8mers and 7mers and confirmed that a similar set of GO-terms was enriched.

Figure 1 – miRNA motif density in expressed target and non-targets 3'UTRs irrespective of AGO1-binding. Putative miRNA motif density in expressed target (red) and non-targets (grey) 3'UTRs irrespective of AGO1-binding. Points represent the predicted miRNA motif density for the top30 CLIP-detected miRNAs. Split violins indicate the predicted miRNA motif density for dinucleotide-shuffled decoy-miRNA sequences. Black bars represent the mean decoy miRNA motif density of 100 individual shuffling experiments. miRNA motif density was normalized to the total target and non-target 3'UTR length. Here, the transcript isoform with the highest isoform percentage has been considered.

Figure 2 – individual 3'UTR miRNA motif density (motif / nt) irrespective of AGO1-binding. Only 25 shuffles are shown because of plotting reasons.

Next, we evaluated the motif density of putative miRNA binding sites in genes associated with each of the enriched GO-term categories presented in Figure 5 of the manuscript. Comparing the putative miRNA motif density of miRNA target gene 3'UTRs with GO-term associated genes that are classified as non-targets shows higher motif density for miRNA target UTRs across the enriched GO-categories (Figure 3). The median 3'UTR length in target genes per GO-category is longer than 3'UTRs in unbound genes for the same categories (Figure 4). We did not see significant correlations between GO-term specific miRNA motif density and median GO-term specific target gene 3'UTR length, between miRNA motif density and GO-term enrichment, or GO-term enrichment and median GO-term specific target gene UTR length (Figure 5).

Figure 3 – miRNA motif density (motif / nt) across individual enriched GO-term categories presented in Figure 5 of the manuscript. For each GO-category, the associated miRNA-targeted and expressed non-targeted genes are compared.

Figure 4 – median 3'UTR length across individual enriched GO-term categories presented in Figure 5 of the manuscript. For each GO-category, the associated miRNA-targeted and expressed non-targeted genes are compared.

Figure 5 – Relationships between (a) GO-term specific miRNA motif density (motif / nt) and median GO-term specific target gene UTR length, (b) between miRNA motif density (motif / nt) and GO-term enrichment, or (c) GO-term enrichment and median GO-term specific target gene UTR length.

[...] In fact, it is a little worrisome that the group of genes showing up-regulation upon AGO1 depletion (“mRNA” group in Figure 5B) does not seem to show particularly stronger enrichment of the same set of GO-terms and may not further support the functional relevance of observations discussed throughout Figure 5 [...]

We see that the very most enriched GO-terms for this group still fall into the same highlighted categories of development, morphogenesis and especially cell-to-cell communication and signaling when sorting Supplemental Data 14 for GO-terms enriched in category mRNA. However, these enrichments are indeed weaker, though the most significant. One main reason for this may be that miRNA targets can visualize direct effects. In contrast, mRNA changes are measured after 72hrs of AGO1 knockdown which allows for secondary effects that might be integrated in the miRNA-targeting effects into a systemic response.

Of course, GO-term analysis only tells us correlation, and I find that the authors are careful in phrasing not to imply any functional importance of miRNAs in the enriched biological processes, but I think this may not be enough for general audience. If there is no strong evidence excluding the possibility that this is a mere result of enriching long 3'UTR genes, the authors should let readers know the potential caveat [...]

Another suggestion is to consider including some of the results presented in Figures 7-13 of the rebuttal letter as supplementary figures and briefly describing them. The information seems useful for the research community.

We have added an additional paragraph at the end of the result section including a new Supplemental Figure 5 as part of Supplemental File 1, which includes the 3'UTR length analysis and putative miRNA motif density analysis. Moreover, we are carefully phrasing the that GO-term enrichments may be circumstantial.

Taken together, we see evidence of enriched putative miRNA motif density for higher order miRNA binding modes (8mers and 7mers) in assigned miRNA-target genes. Gene ontology enrichment analysis on only 7mer and 8mer targeted genes maintain a similar set of enriched GO-terms. We would like to emphasize that putative miRNA motif densities without binding evidence are not a good prior to justify biological relevance (defined non-targets show only marginal decrease in putative miRNA density but are not bound *in vivo*). This argues that the sole presence of a putative motif is not enough to ensure miRNA-targeting. In turn, this highlights the validity of long 3'UTR targeting of miRNAs in S2 cells and suggests that it is not just a 3'UTR length-dependent effect.

REVIEWERS' COMMENTS:

Reviewer #2 (Remarks to the Author):

The authors addressed my concerns sufficiently, and I recommend the manuscript for publication in Nature Communications.